# ART-VITON: MEASUREMENT-GUIDED LATENT DIFFUSION FOR ARTIFACT-FREE VIRTUAL TRY-ON

## ABSTRACT

Virtual try-on (VITON) aims to generate realistic images of a person wearing a target garment, requiring precise garment alignment in try-on regions and faithful preservation of identity and background in non-try-on regions. While latent diffusion models (LDMs) have advanced alignment and detail synthesis, preserving non-try-on regions remains challenging. A common post-hoc strategy directly replaces these regions with original content, but abrupt transitions often produce boundary artifacts. To overcome this, we reformulate VITON as a linear inverse problem and adopt trajectory-aligned solvers that progressively enforce measurement consistency, reducing abrupt changes in non-try-on regions. However, existing solvers still suffer from semantic drift during generation, leading to artifacts. We propose ART-VITON, a measurement-guided diffusion framework that ensures measurement adherence while maintaining artifact-free synthesis. Our method integrates residual prior-based initialization to mitigate training-inference mismatch and artifact-free measurement-guided sampling that combines data consistency, frequency-level correction, and periodic standard denoising. Experiments on VITON-HD, DressCode, and SHHQ-1.0 demonstrate that ART-VITON effectively preserves identity and background, eliminates boundary artifacts, and consistently improves visual fidelity and robustness over state-of-the-art baselines.

## 1 INTRODUCTION

Virtual try-on (VITON) aims to synthesize photorealistic images of a person wearing a desired garment, enabling personalized and immersive online shopping experiences. Given a person image and clothing item, the system must align the garment to the body (try-on regions) while preserving identity (e.g., face, hair) and background (non-try-on regions). Despite progress in generative models, this task remains challenging due to two requirements: precise garment alignment and faithful preservation of non-try-on regions. Various approaches have been proposed to address these challenges (Han et al., 2018; Yu et al., 2019; Yang et al., 2020; Ge et al., 2021; Choi et al., 2021b; Xie et al., 2023; Morelli et al., 2023; Gou et al., 2023; Wang et al., 2024; Kim et al., 2024a; Choi et al., 2024), yet they have primarily focused on garment alignment, leaving the preservation of non-try-on regions largely underexplored.

Early VITON methods (Han et al., 2018; Yu et al., 2019; Yang et al., 2020; Ge et al., 2021) relied on GAN-based two-stage pipelines with garment warping and synthesis networks, which improved alignment but suffered from sensitivity to warping accuracy, instability, and poor generalization due to limited garment-person diversity in existing datasets (Han et al., 2018; Choi et al., 2021b; Morelli et al., 2022). Recent diffusion models (DMs) (Ramesh et al., 2021; Rombach et al., 2022; Podell et al., 2024) address these issues with stable training, broader coverage, and flexible conditioning, achieving higher fidelity and stability. Two-stage approaches (Morelli et al., 2023; Wan et al., 2024) still rely on garment warping, while one-stage approaches (Kim et al., 2024a; Choi et al., 2024) eliminate warping by conditioning on garment features (via LoRA Hu et al. (2022), DreamBooth Ruiz et al. (2023)) or structural signals (via ControlNet Zhang et al. (2023), IP-Adapter Ye et al. (2023)). These advances largely resolve alignment challenges and enable more reliable, detailed synthesis.

Despite significant progress in garment alignment, preserving non-try-on regions has been largely overlooked. Even when models are directly conditioned on such regions, they fail to fully preserve non-try-on areas, resulting in distorted facial features, altered backgrounds, and reduced realism

Figure 1: Comparison of boundary artifacts across methods. We evaluate two criteria: (a) artifact-free outputs and (b) adherence to measurements. StableVITON achieves (a) but fails in (b). Post-hoc replacement enforces (b) but introduces seams, breaking (a). Inverse solvers maintain (b) but suffer semantic drift, degrading (a) over time. ART-VITON satisfies both (a) and (b). Green: success(measurement adherence or artifact-free); red: violations or artifacts. Solid/Dashed boxes show final/intermediate ($t$=835) outputs.

(see Fig. 1, second column; also Appendix Fig. 7). A common strategy (Yang et al., 2020; Xie et al., 2023; Gou et al., 2023) for preserving identity is based on **post-hoc replacement**, where the generated output is projected onto predefined masks or clothing-agnostic maps (Fig. 1, leftmost column) so that non-try-on regions are directly overwritten with original pixels. In this work, we refer to these masks as **measurements**. While intuitive, this approach often introduces **boundary artifacts** at region interfaces, manifesting as color mismatches, lighting inconsistencies, or broken textures (Fig. 1). The root cause is a spatial discontinuity: the generative model evolves freely during inference, unaware of the hard replacement that will occur afterward, resulting in abrupt transition once replacement is applied.

To address the issue of images being generated without completely reflecting measurements, we formulate VITON as a linear inverse problem and integrate existing trajectory-aligned inverse solvers (Chung et al., 2024; Kim et al., 2025) into the latent diffusion model (LDM) sampling process. Compared to post-hoc methods, these solvers progressively guide the latent denoising trajectory, better adhering to measurements and enabling smooth transitions instead of abrupt region replacements. Nevertheless, these solvers can induce semantic inconsistencies between try-on and non-try-on regions during generation, potentially accumulating into boundary artifacts (Fig. 1, fourth column; also Appendix Fig. 8). This limitation highlights the need for a more robust solver that can maintain semantic coherence while satisfying measurements throughout the generation process.

To mitigate semantic drift and enhance visual quality, we propose ART-VITON, a novel latent diffusion inverse solver that enforces measurement consistency during generation, yielding artifact-free synthesis. Our solver incorporates three key components: (i) **data consistency**, preserving semantic coherence and reducing drift, (ii) **frequency-level correction**, restoring high-frequency details lost during pixel-to-latent transition, and (iii) **periodic standard denoising**, leveraging prior knowledge to provide temporal alignment across regions. To avoid instability from direct trajectory manipulation and mitigate training-inference mismatch Lin et al. (2024), a **residual prior** is injected at initialization to maintain both stability and generative diversity. Operating externally without modifying the LDM, our framework is model-agnostic and applicable to diverse VITON pipelines (Fig. 2). Consequently, ART-VITON preserves non-try-on regions, improves garment alignment, eliminates boundary artifacts (Fig. 1), and demonstrates improved results on three benchmark VITON datasets.

## 2 RELATED WORK

### 2.1 IMAGE-BASED VITON METHODS

Early VITON approaches primarily relied on GAN-based two-stage pipelines, where garmets were warped to align with target poses and then integrated into the person image. Pioneering works (Han et al., 2018; Yang et al., 2020) used geometric matching or thin-plate spline transformations, while later methods, including VITON-HD Choi et al. (2021b), HR-VITON Lee et al. (2022), and GP-VTON Xie et al. (2023), extended this framework to high-resolution settings, improving detail preservation. Despite progress, these pipelines remained highly sensitive to warping errors, un-

stable during training, and limited in generalization, while still depending on post-hoc replacement for preserving identity, which introduced boundary artifacts.

Latent diffusion models (LDMs) brought more stable training, better garment fidelity, and controllable synthesis. Two-stage pipelines (e.g., LaDI-VTON Morelli et al. (2023), DCI-VTON Gou et al. (2023), FLDM-VTON Wang et al. (2024), GarDiff Wan et al. (2024)) retain warping modules before diffusion, while one-stage methods bypass warping by encoding garment semantics (e.g., LoRA Hu et al. (2022), Textual Inversion Gal et al. (2023)) or injecting spatial cues through adapters (Zhang et al., 2023; Ye et al., 2023; Hu, 2024; Kingma & Welling, 2022). StableVITON Kim et al. (2024a) strengthens garment–human interaction via a zero cross-attention block in ControlNet Zhang et al. (2023), while Boow-VTON Zhang et al. (2025b) encodes garments with a Parallel U-Net Hu (2024) and integrates them into self-attention to enhance structural representation. DreamPaint Seyfioglu et al. (2023) binds garments to custom tokens using DreamBooth Ruiz et al. (2023). Yet, even with these advances, most LDM-based approaches still rely on post-hoc replacement for non-try-on regions, leaving spatial discontinuity at boundaries unresolved.

## 2.2 DIFFUSION INVERSE SOLVERS

Diffusion inverse solvers aim to integrate measurement constraints into the denoising process. Instead of conditioning on measurements alone, inverse solvers modify the sampling trajectory to align outputs with observations. Early works such as RePaint Lugmayr et al. (2022b) and ILVR Choi et al. (2021a) applied hard projection strategies on pixel-space, while Diffusion Posterior Sampling (DPS) Chung et al. (2023) adjusted sampling trajectories with measurement gradients and Measurement-Constrained Gradient (MCG) Chung et al. (2022) enforced projection onto measurement subspaces. Although these methods improve measurement adherence, they often distort denoising trajectories at high noise levels and accumulate semantic mismatches, producing boundary artifacts. Recent extensions to LDMs attempt to mitigate this. PSLD Rout et al. (2023) extends DPS into the latent domain, Resample Song et al. (2024) reintroduces noise after replacement in an MCG-manner, and TReg Kim et al. (2025) or DreamSampler Kim et al. (2024b) alternate between pixel- and latent-space refinements for stability. While effective in reducing abrupt post-hoc inconsistencies when inverse solvers are applied to VITON, these approaches still fail to maintain smooth semantic coherence between try-on and non-try-on regions, motivating the need for a solver tailored to artifact-free try-on synthesis.

## 3 PRELIMINARIES

### 3.1 LATENT DIFFUSION MODELS

Latent Diffusion Models (LDMs) Rombach et al. (2022) perform the diffusion process in a compressed latent space, improving efficiency while preserving semantics. An input image $\mathbf{x}$ is encoded into a latent code $\mathbf{z}_0 = \mathcal{E}(\mathbf{x})$ via a pre-trained encoder $\mathcal{E}$, which is progressively perturbed into $\mathbf{z}_t$ at timestep $t$ by adding Gaussian noise. At each step, a denoising network $\boldsymbol{\epsilon}_\theta(\mathbf{z}_t, t, \mathbf{c})$ predicts the noise added, conditioned on auxiliary inputs $\mathbf{c}$ (e.g., garments, measurements, or text). Using Tweedie's formula, the posterior latent estimate is:

$$\hat{\mathbf{z}}_0^{(t)} = \frac{1}{\sqrt{\bar{\alpha}_t}} \left( \mathbf{z}_t - \sqrt{1 - \bar{\alpha}_t} \cdot \boldsymbol{\epsilon}_\theta(\mathbf{z}_t, t, \mathbf{c}) \right), \tag{1}$$

where $\bar{\alpha}_t$ is the cumulative noise scale. Based on this, the DDIM Lugmayr et al. (2022a) sampler provides a deterministic update:

$$\mathbf{z}_{t-1} = \sqrt{\bar{\alpha}_{t-1}} \cdot \hat{\mathbf{z}}_0^{(t)} + \sqrt{1 - \bar{\alpha}_{t-1}} \cdot \boldsymbol{\epsilon}_\theta(\mathbf{z}_t, t, \mathbf{c}). \tag{2}$$

These iterative refinements produce high-quality samples while allowing for controllable conditioning.

### 3.2 LINEAR INVERSE PROBLEMS

Many imaging tasks, such as inpainting, super-resolution, and deblurring, can be cast as linear inverse problems, where the observed measurement $\mathbf{y} \in \mathbb{R}^m$ is a partial or degraded version of the

underlying image $\mathbf{x} \in \mathbb{R}^n$. This is generally expressed as:

$$\mathbf{y} = \mathcal{A}\mathbf{x} + \mathbf{n}, \quad \mathbf{n} \sim \mathcal{N}(\mathbf{0}, \sigma^2 \mathbf{I}), \tag{3}$$

where $\mathcal{A} \in \mathbb{R}^{m \times n}$ is a linear operator and $\mathbf{n}$ denotes additive Gaussian noise. The objective is to recover $\mathbf{x}$ that both satisfies the measurements and remains consistent with the natural image distribution. Classical approaches impose explicit priors, while diffusion-based inverse solvers incorporate measurement constraints directly into the denoising process.

## 4 METHOD

### 4.1 REFORMULATING VITON AS AN INVERSE PROBLEM

Virtual try-on requires generating a new garment in try-on regions while preserving identity and background in non-try-on regions. Let $\mathbf{x}$ be the target person image and $\mathbf{y}$ the observed non-try-on regions defined by a clothing-agnostic map (see Fig. 1). This forms a linear inverse problem Eq. 3, where $\mathcal{A}$ is a masking operator. The objective is to reconstruct $\mathbf{x}$ such that *(i)* measurements $\mathbf{y}$ are faithfully preserved, *(ii)* attributes of the reference garment $\mathbf{c}$ are retained, and *(iii)* overall visual coherence is achieved. Since $\mathbf{y}$ is provided to the model as a noise-free conditioning input, it is assumed noise-free, i.e., no noise $\mathbf{n}$ in Eq. 3.

This perspective enables direct incorporation of measurement consistency into the sampling trajectory of LDMs, avoiding reliance on post-hoc replacement. Assuming a well-trained autoencoder $(\mathcal{E}, \mathcal{D})$, the target image $\mathbf{x}$ is reconstructed from the latent vector $\mathbf{z}$ via $\mathbf{x} = \mathcal{D}(\mathbf{z})$ and clean latent estimate $\hat{\mathbf{z}}_0^{(t)}$ in Eq. 1. The conditional distribution then factorizes as:

$$p(\mathbf{x}|\mathbf{y}, \hat{\mathbf{z}}_0^{(t)}) \propto p(\hat{\mathbf{z}}_0^{(t)}|\mathcal{D}(\mathbf{z}), \mathbf{y}) \cdot p(\mathbf{y}|\mathcal{D}(\mathbf{z})), \tag{4}$$

where the first term encourages semantic plausibility (garment fidelity and visual coherence), while the second enforces measurement preservation (non-try-on regions). Standard LDM inference does not explicitly enforce this balance: non-try-on regions evolve freely and are often corrected post-hoc, introducing boundary seams. Existing inverse solvers enforce measurements $\mathbf{y}$ during sampling but often too rigidly, leading to semantic drift and boundary artifacts. We therefore introduce ART-VITON, which directly embeds measurement consistency into the sampling trajectory through two innovations: (a) prior-based initialization and (b) artifact-free measurement-guided sampling.

### 4.2 PRIOR-BASED INITIALIZATION

Eq. 4 defines the VITON posterior as balancing two terms: $p(\hat{\mathbf{z}}_0^{(t)}|\mathcal{D}(\mathbf{z}), \mathbf{y})$ (data consistency) and $p(\mathbf{y}|\mathcal{D}(\mathbf{z}))$ (measurement constraint). For this posterior estimation to be valid, the initial latent $\mathbf{z}_T$ must lie on the noisy data manifold $\mathcal{M}_T$. However, diffusion models suffer from train-test mismatch: training uses $\mathbf{z}_T$ with residual signals at $T{=}999$, while inference commonly starts from pure Gaussian noise at reduced timesteps (e.g., $T{=}981$ in DDIM and VITON baselines (Wan et al., 2024; Kim et al., 2024a)). This mismatch causes $\mathbf{z}_T$ to lie off-manifold, leading to inaccurate posterior terms and error accumulation across denoising steps.

We address this with residual prior-based initialization that places $\mathbf{z}_T$ on $\mathcal{M}_T$ without additional modules. Starting from Gaussian noise $\mathbf{z}_{999}$, we apply one DDPM Ho et al. (2020) denoising step to obtain $\mathbf{z}_{998}$ and use it as $\mathbf{z}_T$ (see Fig. 2 (A)). This ensures both posterior terms in Eq. 4 are computed from on-manifold latents, stabilizing the inverse problem formulation. Table 1 shows consistent improvements across all baselines, confirming that proper initialization directly enhances posterior estimation reliability—the foundation of our inverse solver framework.

### 4.3 ARTIFACT-FREE MEASUREMENT-GUIDED SAMPLING

Naively enforcing measurements during denoising can preserve non-try-on regions but often introduces boundary artifacts, since rigid constraints disrupt semantic continuity. To balance measurement fidelity with artifact-free semantic plausibility, ART-VITON iteratively refines samples to converge toward a latent code $\hat{\mathbf{z}}_0$ that satisfies the measurement constraint, by integrating following complementary techniques, as shown in Fig. 2.

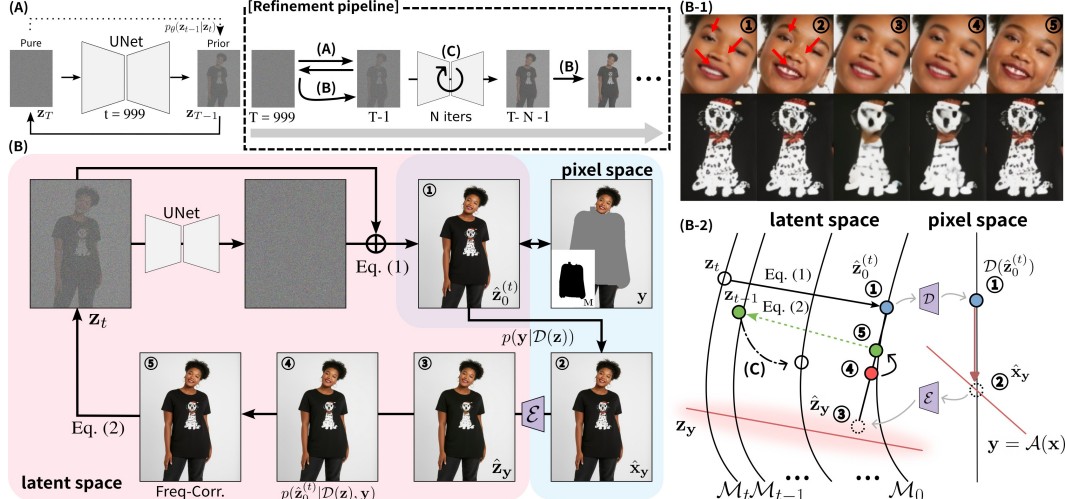

Figure 2: **ART-VITON pipeline.** (A) Prior-based initialization places $\mathbf{z}_T$ on the data manifold, enabling valid posterior sampling for the inverse problem. (B) Artifact-free measurement-guided solver enforces measurements while preserving semantics: ① Tweedie estimation retains garment details but violates measurements in non-try-on regions. ② Hard measurement constraints in pixel space correct preserved regions. ③ VAE re-encoding causes high-frequency loss, which is recovered via ④ data consistency optimization and ⑤ frequency-level correction, detailed in (B-1). (C) Periodic standard denoising realigns the trajectory with noisy data manifolds $\mathcal{M}_t$ for smooth smooth inter-region blending. (B-2) visualizes the complete sampling trajectory.

② **Hard measurement constraint.** At each step, non-try-on regions (in pixel-space) are replaced with ground-truth measurements, directly enforcing $p(\mathbf{y}|\mathcal{D}(\mathbf{z}))$ in Eq. 4 and ensuring faithful identity preservation:

$$\hat{\mathbf{x}}_{\mathbf{y}} = \mathbf{M} \odot \mathbf{y} + (1 - \mathbf{M}) \odot \mathcal{D}(\mathbf{z}), \tag{5}$$

where $\mathbf{M}$ is a binary mask (1 for measurements) and $\mathbf{z}$ is initialized as $\hat{\mathbf{z}}_0^{(t)}$. The updated image $\hat{\mathbf{x}}_{\mathbf{y}}$ is then re-encoded to $\hat{\mathbf{z}}_{\mathbf{y}} = \mathcal{E}(\hat{\mathbf{x}}_{\mathbf{y}})$, which aligns the latent with measurement constraints but may cause information loss, moving $\hat{\mathbf{z}}_{\mathbf{y}}$ away from the semantic trajectory (red line in Fig. 2 (B-2)).

④ **Data consistency.** Hard measurement constraint in ② is insufficient to preserve reference (garment) image attributes, leading to semantic inconsistencies across regions. Thus, focusing on $p(\hat{\mathbf{z}}_0^{(t)}|\mathcal{D}(\mathbf{z}), \mathbf{y})$ in Eq. 4, $\mathbf{z}$ is initialized with $\hat{\mathbf{z}}_{\mathbf{y}}$ and optimized via TReg Kim et al. (2025), i.e., $\hat{\mathbf{z}}_{\mathbf{y}}$ is interpolated toward the reference-informed latent $\hat{\mathbf{z}}_0^{(t)}$ in Eq. 1:

$$\min_{\mathbf{z}} \left\| \frac{\hat{\mathbf{z}}_0^{(t)} - \mathcal{E}(\mathcal{D}(\mathbf{z}))}{2\sigma_{\mathcal{E}}^2} \right\|_2^2, \quad \hat{\mathbf{z}}_0^{(t)}(\bar{\alpha}_{t-1}) = \bar{\alpha}_{t-1}\hat{\mathbf{z}}_{\mathbf{y}} + (1 - \bar{\alpha}_{t-1})\hat{\mathbf{z}}_0^{(t)}, \tag{6}$$

where $\sigma_{\mathcal{E}}$ denotes encoder reconstruction noise and $\bar{\alpha}_{t-1} \in [0, 1]$ controls the interpolation strength.

⑤ **High-frequency correction.** The optimized latent $\hat{\mathbf{z}}_{\mathbf{y}}$ from Eq. 6 closely approximates the true latent $\mathbf{z}_{\mathbf{y}}$ but suffers from high-frequency degradation due to VAE compression—a known limitation typically addressed via retraining (Zhang et al., 2025a; Novitskiy et al., 2025; Almog et al., 2025). Direct interpolation with $\hat{\mathbf{z}}_{\mathbf{y}}$ would propagate this degradation across all regions, causing semantic misalignment between try-on and non-try-on areas.

We address this through frequency-domain correction without retraining. For measurement regions, we construct a corrected latent $\hat{\mathbf{z}}_{\mathbf{y}}' = \hat{\mathbf{z}}_{\mathbf{y}}^{\text{low}} + \hat{\mathbf{z}}_0^{(t),\text{high}}$ via per-channel Fourier transform, fusing low-frequency structure from the optimized $\hat{\mathbf{z}}_{\mathbf{y}}$ with high-frequency details from the reference-informed $\hat{\mathbf{z}}_0^{(t)}$. For masked regions, we directly retain $\hat{\mathbf{z}}_0^{(t)}$, which already contains accurate high-frequency information:

$$\hat{\mathbf{z}}_0^{(t)}(\bar{\alpha}_{t-1}) = \mathbf{M} \odot \left[ \bar{\alpha}_{t-1}\hat{\mathbf{z}}_{\mathbf{y}}' + (1 - \bar{\alpha}_{t-1})\hat{\mathbf{z}}_0^{(t)} \right] + (1 - \mathbf{M}) \odot \hat{\mathbf{z}}_0^{(t)}. \tag{7}$$

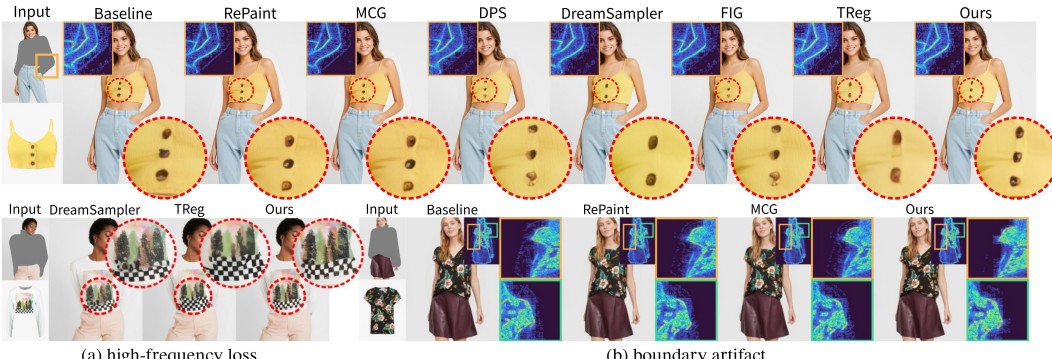

(a) high-frequency loss      (b) boundary artifact

Figure 3: Qualitative comparison across inverse solver categories: (a) Hybrid stochastic methods (DreamSampler, TReg) reduce visible artifacts but lose high-frequency details. (b) Hard constraint (RePaint, MCG) and progressive methods (DPS, FIG) rely on post-hoc replacement, leading to boundary artifacts. Our method mitigates both issues, preserving fine details and enhancing visual coherence across all regions.

This selective refinement preserves semantic alignment across regions while recovering fine-grained details, eliminating artifacts without disturbing garment synthesis.

**(C) Standard denoising.** To avoid instability from repeated measurement-guided corrections, every $N$ steps we apply standard denoising steps, leveraging the diffusion model's inherent ability to harmonize inter-region inconsistencies. This realigns trajectories with the LDM manifold and prevents over-constrained solution, e.g., noisy latent $\mathbf{z}_{t-1}$ is guided to be positioned on the subsequent noisy manifolds (in Fig. 2 (B-2)). Overall, the complete pipeline alternates between measurement-guided updates (A)→(B) and standard denoising (C), following the sequence: (A)→(B)→(C)→(B)→(C)→ . . . , ensuring both measurement consistency and visual fidelity throughout generation.

## 5 EXPERIMENTS

**Dataset.** We evaluate our method on three datasets: VITON-HD (Choi et al., 2021b), Dress-Code (Morelli et al., 2022), and SHHQ-1.0 (Fu et al., 2022). VITON-HD contains $11,647$ training and $2,032$ test pairs of frontal-view female upper-body images ($1024 \times 768$). DressCode includes full-body images with upper/lower/dress items, totaling $15,363$, $8,951$, and $2,947$ pairs, with $1,800$ test pairs per category ($1024 \times 768$); we conduct experiments only on upper-body items. SHHQ-1.0 provides $40K$ high-quality full-body images ($1024 \times 512$); for evaluation, we use the first $2,032$ images, applying VITON-HD preprocessing to generate input conditions.

**Baselines.** We compare against GAN-based (HR-VITON Lee et al. (2022), GP-VTON Xie et al. (2023)) and LDM-based VITON models (LaDI-VTON Morelli et al. (2023), DCI-VTON Gou et al. (2023), GarDiff Wan et al. (2024), StableVITON Kim et al. (2024a), IDM-VTON Choi et al. (2024), OOTDiffusion Xu et al. (2025), ITA-MDT Hong et al. (2025)). We also benchmark inverse solvers, categorized as: hard constraint (RePaint Lugmayr et al. (2022b), MCG Chung et al. (2022)), progressive update (DPS Chung et al. (2023), FIG Yan et al. (2025)), and hybrid stochastic (Dream-Sampler Kim et al. (2024b), TReg Kim et al. (2025)). Unless otherwise noted, all comparisons use post-hoc replacement, which is also required for hard constraint and progressive update solvers as they fail to fully preserve measurements. See Appendices A.2 and A.3 for details of VITON and inverse solvers.

**Evaluation metric.** We evaluate performance under two settings: paired, where the model reconstructs the original clothing, and unpaired, where the clothing is replaced. In the paired setting, we report PSNR and SSIM for pixel fidelity and structural consistency, and LPIPS for perceptual similarity. In the unpaired setting, we adopt FID to measure visual realism and global distributional coherence, and KID to assess sample diversity.

### 5.1 IMPACT OF PRIOR-BASED INITIALIZATION

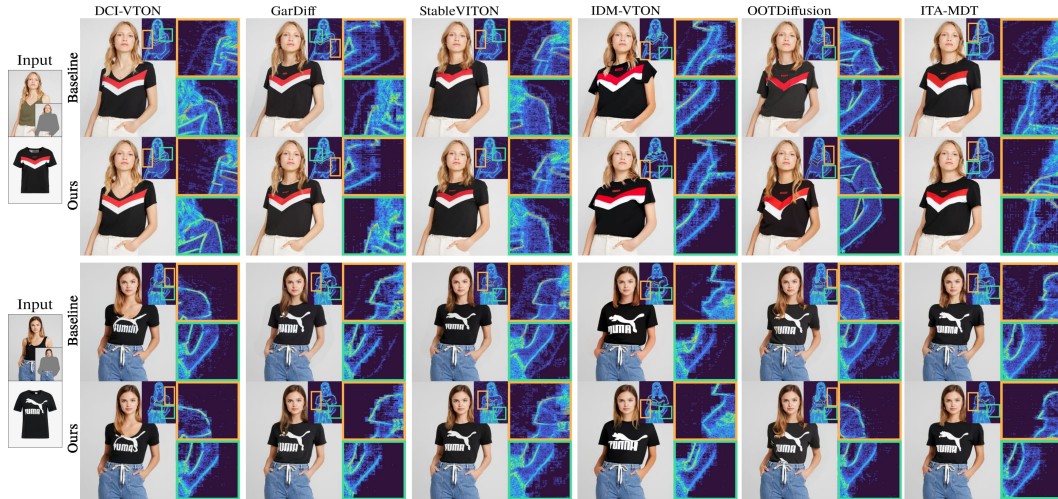

Figure 4: Qualitative results on VITON-HD. Gradient magnitude heatmaps reveal spatial disconti-nuities at region boundaries (necklines, sleeves, waistlines) in baseline models. Our method sub-stantially reduces these artifacts while preserving garment details (patterns, textures, logos).

Our prior-based initialization miti-gates the train-test mismatch and consistently improves performance across all architectures (Table 1). By default, all baselines start denois-ing at $T$=981: DCI-VTON overlays warped garments from its module, GarDiff initializes with pure Gaus-sian noise, and StableVITON uses noisy real images. Since StableVI-

| Model | SSIM ↑ | PSNR ↑ | LPIPS ↓ | FID ↓ | KID ↓ |
|---|---|---|---|---|---|
| DCI-VTON Gou et al. (2023) | 0.8607 | 23.6629 | 0.0852 | 12.6386 | 0.0014 |
| **+ Prior @ T=999** | **0.8880** | **24.1447** | **0.0782** | **11.4713** | **0.0011** |
| GarDiff Wan et al. (2024) | 0.8062 | 21.1075 | 0.1016 | 11.7048 | 0.0061 |
| **+ Prior @ T=999** | **0.8448** | **21.8611** | **0.0864** | **10.5322** | **0.0034** |
| StableVITON Kim et al. (2024a) | 0.8550 | 23.1214 | 0.0835 | 10.8716 | 0.0022 |
| **+ Prior @ T=999** | **0.8552** | **23.1475** | **0.0833** | **10.4362** | **0.0014** |

Table 1: Effect of prior-based initialization at $T$=999 across baseline models on VITON-HD. Our method consistently improves all metrics regardless of architecture.

TON's initialization is tailored for unpaired settings, we replaced $\mathbf{z}_T$ with pure noise for fair paired comparisons. Adjusting starting timestep $T$=999 alone already boosts performance, particularly for StableVITON (paired) and DCI-VTON. In unpaired settings, our residual prior-based initial-ization better fills masked regions with plausible structure, yielding sharper and more consistent garments, especially for StableVITON. GarDiff also shows notable gains, demonstrating the broad utility across architectures of our approach.

| Method | SSIM ↑ | PSNR ↑ | LPIPS ↓ | FID ↓ | KID ↓ | FLOPs (T) ↓ | Inf. (s) ↓ | Mem. (GB) ↓ |
|---|---|---|---|---|---|---|---|---|
| StableVITON (baseline) | 0.8839 | 23.5965 | 0.0757 | 9.8694 | 0.0016 | 86.178 | 9.117 | 7.66 |
| RePaint Lugmayr et al. (2022b) | 0.8856 | 23.6635 | 0.0752 | 10.0829 | 0.0018 | 87.864 | 9.241 | 7.66 |
| MCG Chung et al. (2022) | 0.8855 | 23.6641 | 0.0752 | 10.085 | 0.0015 | 259.87 | 13.518 | 10.38 |
| DPS Chung et al. (2023) | 0.8851 | 23.6390 | 0.0749 | 9.9425 | 0.0014 | 259.87 | 13.573 | 10.38 |
| DreamSampler Kim et al. (2024b) | 0.8904 | **23.8984** | 0.0771 | 10.5143 | 0.0018 | 267.22 | 25.463 | 7.66 |
| FIG Yan et al. (2025) | 0.8851 | 23.6390 | 0.0749 | 9.9427 | 0.0014 | 259.87 | 13.389 | 10.38 |
| TReg Kim et al. (2025) | **0.8909** | 23.8205 | 0.0844 | 11.7467 | 0.0024 | 130.63 | 11.382 | 7.66 |
| Ours | 0.8859 | 23.7027 | **0.0746** | **9.7669** | **0.0009** | 130.63 | 12.101 | 7.66 |

Table 2: Comparison of StableVITON with existing inverse solvers on VITON-HD. All methods use identical (A) initialization and (C) denoising steps from Fig. 2; only measurement-guided sampling (B) differs. Red cells: degradation vs. baseline. **Bold**: best, underline: second-best. Inf.: inference time (s) per image; Mem.: memory usage (GB). Existing solvers exhibit trade-offs—improving paired metrics at the cost of unpaired performance—while ours achieves balanced improvements.

## 5.2 COMPARISON WITH EXISTING INVERSE SOLVERS

Table 2 compares our method with existing inverse solver categories—hard constraint, progressive update, and hybrid stochastic. Prior solvers face a trade-off: they improve paired metrics (SSIM, PSNR) but degrade unpaired performance (FID, KID), falling below baseline in perceptual qual-ity. In contrast, ART-VITON achieves balanced improvements across all metrics. Hard constraint methods (RePaint (Lugmayr et al., 2022b), MCG (Chung et al., 2022)) enforce measurements in

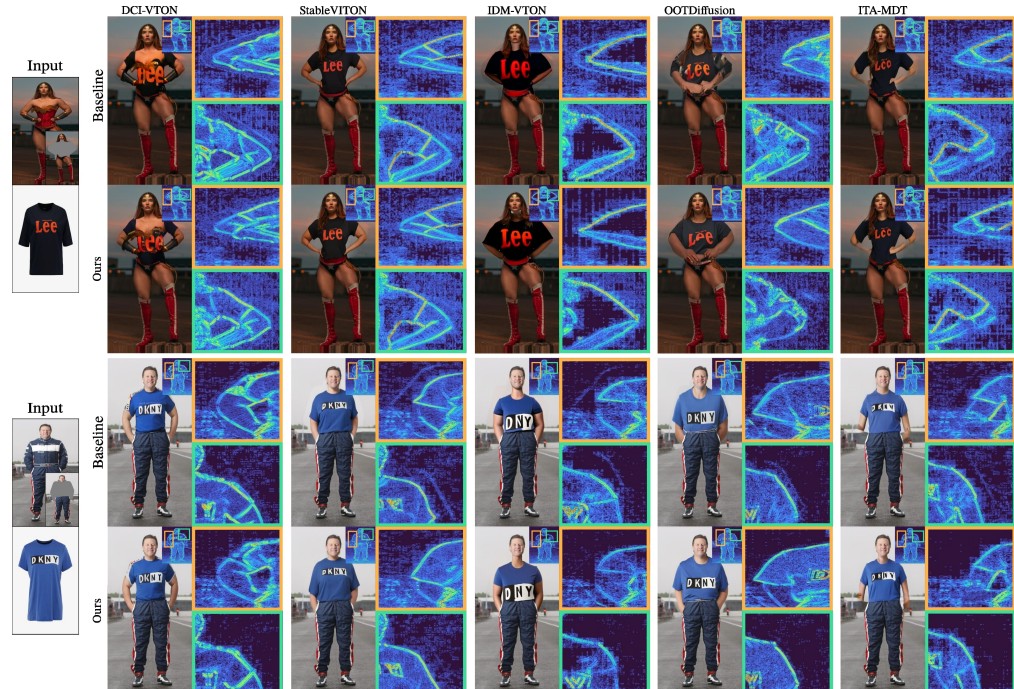

Figure 5: Cross-domain results on SHHQ-1.0. Models trained on VITON-HD are evaluated on in-the-wild images. Our method effectively mitigates artifacts across all baselines despite diverse poses, lighting conditions, and garment styles, demonstrating strong generalization.

latent space but fail to fully satisfy them. Despite aggressive enforcement, incomplete measurement satisfaction forces reliance on post-hoc replacement, causing abrupt transitions (Fig. 3b) that slightly improve paired metrics but degrade unpaired performance. Progressive update methods (DPS (Chung et al., 2023), FIG (Yan et al., 2025)) optimize more smoothly but still leave spatial discontinuities, requiring post-hoc correction and resulting in unpaired metrics below baseline.

Hybrid stochastic solvers (DreamSampler (Kim et al., 2024b), TReg (Kim et al., 2025)) inject stochastic noise to soften transitions, reducing visible artifacts. However, stochastic perturbations degrade unpaired metrics, and VAE re-encoding loses high-frequency details, further harming perceptual quality (LPIPS, Fig. 3a). Fig. 3 (top row) shows that these limitations appear consistently across all solver categories. DreamSampler also incurs high inference cost, while gradient-based methods require substantial memory. Our method maintains semantic alignment and preserves high-frequency details throughout generation, achieving both measurement satisfaction and artifact-mitigated synthesis at reasonable computational cost.

## 5.3 COMPARISON WITH VITON BASELINES

**VITON-HD results.** Table 3 shows that our method consistently improves all baselines in the in-domain setting (VITON-HD/VITON-HD). It boosts paired metrics (SSIM, PSNR, LPIPS) by reducing boundary artifacts and preserving high-frequency details, while also improving unpaired metrics (FID, KID) through better semantic alignment. Fig. 4 illustrates this improvement: baseline models exhibit boundary artifacts in gradient heatmaps around necklines, sleeves, and waistlines, whereas our method removes these discontinuities and preserves fine garment details such as patterns, textures, logos, and text. Comprehensive results are presented in See Fig. 12.

**Cross-Domain generalization.** The large domain gap between studio-quality and in-the-wild images poses challenges, yet our method improves performance (Table 3, right columns). It reduces artifacts across diverse baselines (DCI-VTON, StableVITON, IDM-VTON, OOTDiffusion, ITA-MDT), enhancing visual coherence under varying poses, lighting, and garment styles. Baselines with moderate artifacts (e.g., DCI-VTON, StableVITON) achieve near-artifact-free results, while those with severe artifacts (OOTDiffusion) improve noticeably but retain minor imperfections

| Dataset(train/test) | VITON-HD / VITON-HD | | | | | VITON-HD / SHHQ | |
|---|---|---|---|---|---|---|---|
| Method | SSIM ↑ | PSNR ↑ | LPIPS ↓ | FID ↓ | KID ↓ | FID ↓ | KID ↓ |
| HR-VITON Lee et al. (2022) | 0.8710 | 22.3368 | 0.0986 | 11.7301 | 0.3926 | 36.2665 | 0.0184 |
| GP-VTON Xie et al. (2023) | 0.8718 | 23.6485 | 0.0838 | 12.0564 | 0.0029 | – | – |
| LaDI-VTON Morelli et al. (2023) | 0.8779 | 22.7451 | 0.0876 | 10.5203 | **0.0004** | 22.2632 | 0.0045 |
| DCI-VTON* Gou et al. (2023) | 0.8871 | 24.1413 | 0.0782 | 11.3634 | 0.0012 | 21.2350 | 0.0055 |
| DCI-VTON† | 0.8908 | 24.5180 | 0.0746 | 10.9724 | 0.0022 | 21.3168 | 0.0048 |
| DCI-VTON (Ours) | **0.8946** | **24.6903** | **0.0722** | 10.5408 | 0.0005 | **21.1485** | **0.0040** |
| GarDiff* Wan et al. (2024) | 0.8418 | 21.7263 | 0.0895 | 10.5858 | 0.0042 | – | – |
| GarDiff† | 0.8413 | 21.8914 | 0.0912 | 11.2894 | 0.0047 | – | – |
| GarDiff (Ours) | 0.8463 | 21.9647 | 0.0866 | 10.3414 | 0.0036 | – | – |
| StableVITON* Kim et al. (2024a) | 0.8839 | 23.5965 | 0.0757 | 9.8694 | 0.0016 | 22.7463 | 0.0066 |
| StableVITON† | 0.8832 | 23.5586 | 0.0772 | 9.9520 | 0.0017 | 22.9052 | 0.0061 |
| StableVITON (Ours) | 0.8859 | 23.7027 | 0.0746 | 9.7669 | 0.0009 | 22.5525 | **0.0040** |
| IDM-VTON* Choi et al. (2024) | 0.8440 | 20.1067 | 0.1193 | 11.5482 | 0.0050 | 27.1165 | 0.0125 |
| IDM-VTON† | 0.8441 | 20.1238 | 0.1201 | 12.1464 | 0.0058 | 26.4503 | 0.0105 |
| IDM-VTON (Ours) | 0.8477 | 20.4514 | 0.1183 | 11.5364 | 0.0048 | 24.3281 | 0.0086 |
| OOTDiffusion* Xu et al. (2025) | 0.8601 | 20.5861 | 0.0977 | 10.2954 | 0.0013 | 22.2065 | 0.0056 |
| OOTDiffusion† | 0.8564 | 20.6652 | 0.0968 | 10.3767 | 0.0022 | 22.2288 | 0.0058 |
| OOTDiffusion (Ours) | 0.8583 | 21.3201 | 0.0953 | **9.5175** | 0.0009 | 23.0877 | 0.0052 |
| ITA-MDT* Hong et al. (2025) | 0.8760 | 23.5203 | 0.0764 | 9.7530 | 0.0025 | 23.0280 | 0.0073 |
| ITA-MDT† | 0.8813 | 23.5662 | 0.0748 | 10.1198 | 0.0029 | 22.3726 | 0.0072 |
| ITA-MDT (Ours) | 0.8820 | 23.5735 | 0.0737 | 9.5977 | 0.0024 | 22.0231 | 0.0059 |

Table 3: Quantitative comparison on VITON-HD and cross-domain evaluation on SHHQ-1.0. Left columns show same-domain results (VITON-HD/VITON-HD), right columns show generalization capability (VITON-HD/SHHQ-1.0). ∗ indicates post-hoc replacement; † indicates post-hoc Poisson blending. Our method, applied without architectural modifications, consistently improves all baseline models across both in-domain and cross-domain settings.

(Fig. 5). Overall, all methods benefit from our approach, producing results better suited for practical use. GP-VTON and GarDiff were excluded due to dataset-specific preprocessing.

**Comparison with Poisson blending.** We evaluate Poisson blending (Pérez et al., 2023), a gradient-domain technique used in CAT-DM (Zeng et al., 2024), against post-hoc replacement (Table **??**). Results are inconsistent: most baselines show degraded unpaired metrics (FID, KID), and some also worsen in paired metrics. This occurs because Poisson blending enforces gradient continuity at boundaries but cannot fix intensity or texture mismatches, often leaving distortions at junctions (Fig. 9). In contrast, our method addresses these issues during generation, maintaining semantic alignment and restoring high-frequency details, yielding visually coherent. This indicates that effective artifact mitigation requires intervention during generation, not post-hoc correction.

**DressCode results.** On DressCode upper-body, our method consistently improves performance and reduces boundary artifacts observed in prior approaches (Table 4, Fig. 15). Existing methods struggle with complex poses and long garments: GP-VTON produces severe distortions, LaDI-VTON suffers from texture degrada-

| Method | SSIM ↑ | PSNR ↑ | LPIPS ↓ | FID ↓ | KID ↓ |
|---|---|---|---|---|---|
| GP-VTON | 0.8876 | 26.5946 | 0.0864 | 15.0994 | 0.0022 |
| LaDI-VTON | 0.9298 | 24.8196 | 0.0498 | 14.5299 | 0.0013 |
| StableVITON | 0.9366 | 26.5536 | 0.0365 | 13.0582 | 0.0015 |
| StableVITON (Ours) | **0.9377** | **26.7143** | **0.0361** | **13.0083** | **0.0009** |

Table 4: Quantitative evaluation on DressCode upper-body. Our method consistently improves all metrics, showing robust performance in full-body scenarios.

tion, and baseline StableVITON exhibits boundary seams. In contrast, StableVITON enhanced with our solver suppresses these artifacts, achieving visually coherent results across challenging cases.

## 5.4 ABLATION STUDY

**Initialization strategy analysis.** Our Prior (DDPM) initialization achieves balanced gains across both paired and unpaired metrics (Table 5). Injecting data into $\mathbf{z}_T$ boosts paired metrics (SSIM, PSNR, LPIPS) by preserving structure, while semantic alignment benefits unpaired metrics (FID, KID). Alternative strategies reveal clear trade-offs: *Pure* lacks real data, lowering paired metrics; *Unmasked* replaces measurement regions with noisy observations, misaligning semantics and degrading FID/KID; *Offset noise* adds global correlated noise to expand brightness range, which pre-

| $\mathbf{z}_T$ @ T=999 | SSIM ↑ | PSNR ↑ | LPIPS ↓ | FID ↓ | KID ↓ |
|---|---|---|---|---|---|
| Pure | 0.855 | 23.1214 | 0.0835 | 10.4349 | 0.0014 |
| Pure (51 step) | 0.855 | 23.1363 | 0.0834 | 10.4451 | **0.0012** |
| Unmasked | **0.8566** | **23.2551** | 0.0834 | 10.6985 | 0.0016 |
| Offset noise | 0.8425 | 22.1414 | 0.0962 | **10.3335** | 0.0015 |
| Prior (DDIM) | 0.855 | 23.1299 | 0.0835 | 10.4631 | 0.0015 |
| Prior (DDPM) | 0.8552 | 23.1475 | **0.0833** | 10.4362 | 0.0014 |

| Method | SSIM ↑ | PSNR ↑ | LPIPS ↓ | FID ↓ | KID ↓ |
|---|---|---|---|---|---|
| Pure | 0.8530 | 23.0727 | 0.0843 | 10.7491 | 0.0018 |
| + (A) Prior-based | 0.8552 | 23.1475 | 0.0833 | 10.4362 | 0.0014 |
| + ② Hard measure. | 0.8677 | 22.5991 | 0.1623 | 20.1817 | 0.0088 |
| + ④ Data consist. | 0.8855 | 23.4532 | 0.1064 | 14.0034 | 0.0029 |
| + ⑤ Freq-Corr. | 0.8861 | 23.7138 | 0.0749 | 9.8644 | 0.0013 |
| + (C) Std. denoising | **0.8859** | **23.7027** | **0.0746** | **9.7669** | **0.0009** |

Table 5: Quantitative comparison of $\mathbf{z}_T$ configurations at $T$=999 on StableVITON (VITON-HD). Prior (DDPM) achieves a good balance, showing strong performance across all metrics.

Table 6: Ablation study on StableVITON (VITON-HD). Incrementally adding each component of our method leads to consistent improvements, confirming their complementary roles.

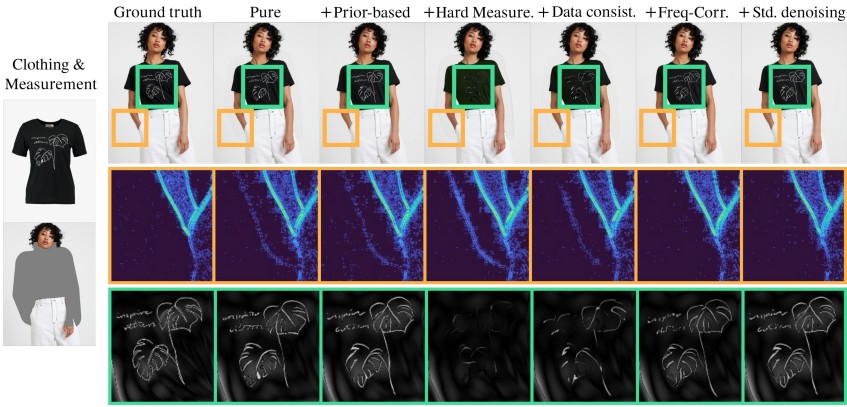

Figure 6: Ablation study of pipeline components. Direct measurement enforcement increases artifacts, while subsequent additions (data consistency, frequency correction, and periodic denoising) progressively reduce them, yielding artifact-free and coherent results.

serves semantic alignment and improves FID/KID but lacks real data, leading to poor paired metrics; *Prior (DDIM)* reduces diversity due to deterministic sampling. In contrast, *Prior (DDPM)* injects minimal semantic structure into initialization, aligning masked and measured regions while retaining diversity, yielding the most balanced performance at $T$=999.

**Component contribution.** We further assess each module's role. (A) Prior-based initialization stabilizes trajectories and improves overall quality (Table 6). ② Direct measurement enforcement guarantees constraint satisfaction but introduces severe boundary artifacts, showing the need for semantic alignment (Fig. 6). ④ Data consistency mitigates residual artifacts but only partially. ⑤ Frequency correction recovers high-frequency details lost in VAE encoding, improving semantic alignment across regions. (C) Periodic standard denoising leverages LDM priors for harmonization, stabilizing trajectories, and enhancing coherence. Together, these results confirm that each component is complementary, and their integration is essential for artifact-free, coherent synthesis.

# 6 CONCLUSION

We propose ART-VITON, a model-agnostic framework that addresses boundary artifacts in virtual try-on. By reformulating VITON as a linear inverse problem and using measurement-guided diffusion sampling, it preserves non-try-on regions and maintains garment alignment. Key innovations include prior-based initialization to reduce training-inference mismatch and artifact-free sampling via data consistency, frequency-level correction, and standard denoising. Experiments show improved boundary coherence and high-frequency detail. ART-VITON delivers accurate, artifact-free virtual try-on, providing users with a realistic and trustworthy preview of fit and style. Extending ART-VITON to other editing tasks is promising, as artifact-free design generalizes beyond VITON.

**Limitations.** While ART-VITON effectively reduces boundary artifacts, its performance depends on the quality of the underlying baseline. Models with severe inherent artifacts—such as major semantic drift or measurement violations—benefit from our method but may retain minor imperfections. Combining our solver with stronger baselines could further improve results. Nevertheless, the model-agnostic design ensures consistent gains across all baselines.

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

# A APPENDIX

## A.1 USE OF LARGE LANGUAGE MODELS

We used a large language model OpenAI (2025) solely to improve the clarity and readability of the manuscript (e.g., grammar and phrasing). The model did not contribute to research ideation, methodology, or analysis, and the authors take full responsibility for all contents.

## A.2 IMPLEMENTATION DETAILS OF BASELINES

We evaluate our method across diverse baseline models with varying architectures and configurations. Our experiments include DCI-VTON Gou et al. (2023) and StableVITON Kim et al. (2024a), both built on Stable Diffusion Rombach et al. (2022) v1.4; GarDiff Wan et al. (2024), which is based on SD v2.1; IDM-VTON Choi et al. (2024) leveraging SDXL Podell et al. (2024) inpainting; OOT-Diffusion Xu et al. (2025) using SD v1.5; and ITA-MDT Hong et al. (2025) built upon the Masked Diffusion Transformer Gao et al. (2023). We utilize publicly available pretrained checkpoints for all baselines, with the exception of StableVITON, which we train on the DressCode Morelli et al. (2022) dataset focusing on upper-body items for consistency. The original baseline configurations vary significantly in their sampling strategies. DCI-VTON, GarDiff, StableVITON, IDM-VTON, and OOTDiffusion begin sampling from timestep $T{=}981$, while ITA-MDT starts at $T{=}999$. Initial latent construction also differs across methods: DCI-VTON overlays warped garments from its warping module, StableVITON uses noisy real images, while GarDiff, IDM-VTON, OOTDiffusion, and ITA-MDT initialize with pure Gaussian noise. Classifier-free guidance Ho & Salimans (2022) scales range from $1.0$ (DCI-VTON, GarDiff, StableVITON) to $2.0$ (IDM-VTON, OOTDiffusion, ITA-MDT) and $7.5$ (LaDI-VTON).

For consistent evaluation, we standardize the inference protocol by applying prior-based initialization to all baselines and employing the DDIM Lugmayr et al. (2022a) sampler with $50$ timesteps. This unified setup enables fair comparison while demonstrating the model-agnostic nature of our approach across different architectural paradigms. For inverse solvers, all methods are adapted to the latent diffusion framework, sharing the same (A) initialization and (C) standard denoising steps ($N = 2$), differing only in the (B) measurement-guided sampling component.

## A.3 INVERSE SOLVER FORMULATION

We classify inverse solvers into three types: hard constraints (RePaint Lugmayr et al. (2022b), MCG Chung et al. (2022)), progressive updates (DPS Chung et al. (2023), FIG Yan et al. (2025)), and hybrid stochastic methods (DreamSampler Kim et al. (2024b), TReg Kim et al. (2025)). Hard constraints induce semantic drift between regions due to strong measurement enforcement, directly causing boundary artifacts. Progressive updates maintain stable optimization and produce minimal artifacts. However, both hard constraints and progressive updates operate in latent space, failing to fully satisfy measurements (Fig. 10). To address this, we apply post-hoc replacement, which can still cause boundary artifacts due to semantic mismatch and spatial discontinuities. Hybrid stochastic methods enforce measurement constraints in pixel space and inject stochastic noise to harmonize regions, reducing artifacts. Nevertheless, persistent semantic drift still leads to artifact formation.

We formulate virtual try-on as an inverse problem and integrate various solver strategies into the latent diffusion sampling process. This section presents the mathematical foundations and implementation details of each approach. We denote the measurement mask as $\mathbf{M}$ and the target measurement as $\mathbf{y}$. The bar notation indicates resizing to match the latent code resolution. Specifically, $\bar{\mathbf{M}}$ denotes the measurement mask with value $1$ in the resized measurement region, and $\bar{\mathbf{y}}$ represents the resized target measurement. A comparison with the inverse solvers is shown in Fig. 11.

**DDIM sampling Lugmayr et al. (2022a).** The deterministic DDIM sampling forms the basis for all inverse solvers. Given a noisy latent $\mathbf{z}_t$ at timestep $t$, we first estimate the clean latent using Tweedie's formula:

$$\hat{\mathbf{z}}_0^{(t)} = \frac{1}{\sqrt{\bar{\alpha}_t}} \left( \mathbf{z}_t - \sqrt{1 - \bar{\alpha}_t} \cdot \boldsymbol{\epsilon}_\theta(\mathbf{z}_t, t, \mathbf{c}) \right). \tag{8}$$

The denoising step then updates the latent to timestep $t - 1$:

$$\mathbf{z}_{t-1} = \sqrt{\bar{\alpha}_{t-1}} \hat{\mathbf{z}}_0^{(t)} + \sqrt{1 - \bar{\alpha}_{t-1}} \boldsymbol{\epsilon}_\theta(\mathbf{z}_t, t, \mathbf{c}). \tag{9}$$

### A.3.1 HARD MEASUREMENT METHODS

These methods enforce measurement consistency through direct projection or replacement in the latent space.

**RePaint Lugmayr et al. (2022b).** This approach replaces the measurement region with noisy observations at each denoising step. We omit the resampling strategy proposed in Repaint as it is too time-consuming:

$$\bar{\mathbf{y}}_{t-1} \sim \mathcal{N}(\sqrt{\bar{\alpha}_{t-1}} \bar{\mathbf{y}}, (1 - \bar{\alpha}_{t-1})\mathbf{I}), \tag{10}$$

$$\mathbf{z}'_{t-1} = \bar{\mathbf{M}} \odot \bar{\mathbf{y}}_{t-1} + (1 - \bar{\mathbf{M}}) \odot \mathbf{z}_{t-1}. \tag{11}$$

**MCG (Manifold-Constrained Gradient) Chung et al. (2022).** This method combines gradient-based optimization with hard projection:

$$\mathbf{z}'_{t-1} = \mathbf{z}_{t-1} - \gamma \nabla_{\mathbf{z}_t} \|\bar{\mathbf{y}} - \bar{\mathbf{M}} \odot \hat{\mathbf{z}}_0^{(t)}\|_2^2, \tag{12}$$

$$\mathbf{z}''_{t-1} = \bar{\mathbf{M}} \odot \bar{\mathbf{y}}_{t-1} + (1 - \bar{\mathbf{M}}) \odot \mathbf{z}'_{t-1}, \tag{13}$$

where $\gamma$ is the gradient step size, which we set to 1.

### A.3.2 PROGRESSIVE UPDATE METHODS

These methods guide the sampling trajectory iteratively through gradient updates without relying on hard measurement constraints.

**DPS (Diffusion Posterior Sampling) Chung et al. (2023).** DPS adjusts the sampling trajectory via measurement consistency gradients computed in the Tweedie space:

$$\mathbf{z}'_{t-1} = \mathbf{z}_{t-1} - \gamma \nabla_{\mathbf{z}_t} \|\bar{\mathbf{y}} - \bar{\mathbf{M}} \odot \hat{\mathbf{z}}_0^{(t)}\|_2^2, \tag{14}$$

where we set $\gamma = 1$.

**FIG (Flow with Interpolant Guidance) Yan et al. (2025).** By operating directly on the noisy latent, FIG performs gradient updates along the diffusion trajectory, preserving stability and sample diversity, whereas Tweedie-space optimization is more precise but incurs higher computational cost and reduces diversity.

$$\mathbf{z}'_{t-1} = \mathbf{z}_{t-1} - \gamma \nabla_{\mathbf{z}_{t-1}} \|\bar{\mathbf{y}}_{t-1} - \bar{\mathbf{M}} \odot \mathbf{z}_{t-1}\|_2^2, \tag{15}$$

with $\gamma = 1$.

### A.3.3 HYBRID STOCHASTIC METHODS

These approaches combine deterministic updates with stochastic noise injection, where the degree of stochasticity is controlled through $\eta\beta_t$, to balance measurement consistency and generation diversity.

$$\tilde{\boldsymbol{\epsilon}}_t := \frac{\sqrt{1 - \bar{\alpha}_{t-1} - \eta^2\beta_t^2} \cdot \boldsymbol{\epsilon}_\theta + \eta\beta_t \cdot \boldsymbol{\epsilon}}{\sqrt{1 - \bar{\alpha}_{t-1}}}, \quad \boldsymbol{\epsilon} \sim \mathcal{N}(0, \mathbf{I}), \tag{16}$$

where $\eta$ controls the noise level and $\beta_t$ is the noise schedule. The pixel-space optimization is solved via conjugate gradient (CG) with a regularization coefficient $\lambda$ of $1e-4$:

**DreamSampler Kim et al. (2024b).** DreamSampler integrates pixel-space and latent-space optimization to guide the diffusion sampling trajectory while maintaining measurement consistency. Let $\varnothing$ denote a null embedding, as introduced in the classifier-free guidance (CFG) framework, used to perform latent optimization without conditioning information. In the final latent update, the stochastic noise term $\tilde{\boldsymbol{\epsilon}}_t$ is set by $\eta\beta_t = \sqrt{\bar{\alpha}_t(1 - \bar{\alpha}_{t-1})}$, controlling the amount of injected noise to balance diversity and trajectory stability.

$$\hat{\mathbf{z}}_{0,\varnothing}^{(t)} = \frac{1}{\sqrt{\bar{\alpha}_t}}\left(\mathbf{z}_t - \sqrt{1 - \bar{\alpha}_t} \cdot \boldsymbol{\epsilon}_\theta(\mathbf{z}_t, t, \varnothing)\right), \tag{17}$$

$$\hat{\mathbf{x}}_{\mathbf{y},\varnothing} = \arg\min_{\mathbf{x}_\varnothing}\left(\|\mathbf{y} - \bar{\mathbf{M}} \odot \mathbf{x}_\varnothing\|_2^2 + \lambda\|\mathbf{x}_\varnothing - \mathcal{D}(\hat{\mathbf{z}}_{0,\varnothing}^{(t)})\|_2^2\right), \quad \hat{\mathbf{z}}_{\mathbf{y},\varnothing} = \mathcal{E}(\hat{\mathbf{x}}_{\mathbf{y},\varnothing}), \tag{18}$$

$$\hat{\mathbf{z}}_0^{(t)}(\bar{\alpha}_{t-1}) = \bar{\alpha}_{t-1}\hat{\mathbf{z}}_{\mathbf{y},\varnothing} + (1 - \bar{\alpha}_{t-1})\hat{\mathbf{z}}_{0,\varnothing}^{(t)}, \tag{19}$$

$$\hat{\mathbf{z}}_0^{(t)}(\bar{\alpha}_t, \bar{\alpha}_{t-1}) = \bar{\mathbf{M}} \odot \hat{\mathbf{z}}_0^{(t)}(\bar{\alpha}_{t-1}) + (1 - \bar{\mathbf{M}}) \odot (\bar{\alpha}_t\hat{\mathbf{z}}_0^{(t)} + (1 - \bar{\alpha}_t)\hat{\mathbf{z}}_0^{(t)}(\bar{\alpha}_{t-1})), \tag{20}$$

$$\mathbf{z}_{t-1}' = \sqrt{\bar{\alpha}_{t-1}}\hat{\mathbf{z}}_0^{(t)}(\bar{\alpha}_t, \bar{\alpha}_{t-1}) + \sqrt{1 - \bar{\alpha}_{t-1}}\tilde{\boldsymbol{\epsilon}}_t, \tag{21}$$

where $\mathcal{E}$ and $\mathcal{D}$ denote encoder and decoder, and $\lambda$ balances data fidelity.

**TReg Kim et al. (2025).** TReg performs the hybrid approach by performing optimization directly in pixel space with latent regularization. It solves a regularized inverse problem where the measurement operator $\bar{\mathbf{M}}$ enforces constraints, while the regularization term maintains semantic coherence via the diffusion prior. In the stochastic update, the noise parameter is set as $\eta\beta_t = \sqrt{\bar{\alpha}_{t-1}(1 - \bar{\alpha}_{t-1})}$, following a noise schedule distinct from DreamSampler.

$$\hat{\mathbf{x}}_{\mathbf{y}} = \arg\min_{\mathbf{x}}\left(\|\mathbf{y} - \bar{\mathbf{M}} \odot \mathbf{x}\|_2^2 + \lambda\|\mathbf{x} - \mathcal{D}(\hat{\mathbf{z}}_0^{(t)})\|_2^2\right), \quad \hat{\mathbf{z}}_{\mathbf{y}} = \mathcal{E}(\hat{\mathbf{x}}_{\mathbf{y}}), \tag{22}$$

$$\hat{\mathbf{z}}_0^{(t)}(\bar{\alpha}_{t-1}) = \bar{\alpha}_{t-1}\hat{\mathbf{z}}_{\mathbf{y}} + (1 - \bar{\alpha}_{t-1})\hat{\mathbf{z}}_0^{(t)}, \tag{23}$$

$$\mathbf{z}_{t-1}' = \sqrt{\bar{\alpha}_{t-1}}\hat{\mathbf{z}}_0^{(t)}(\bar{\alpha}_{t-1}) + \sqrt{1 - \bar{\alpha}_{t-1}}\tilde{\boldsymbol{\epsilon}}_t. \tag{24}$$

## A.4 ADDITIONAL ABLATION STUDY

We conduct ablation studies on two key hyperparameters: the interpolation weights in data consistency optimization (Eq. 6) and the frequency of standard denoising steps (component (C) in Fig. 2).

**Interpolation weights in data consistency.** Eq. 6 performs data consistency optimization by finding the latent $\mathbf{z}$ that balances between the current optimized latent $\hat{\mathbf{z}}_{\mathbf{y}}$ and the reference-informed latent $\hat{\mathbf{z}}_0^{(t)}$. This can be formulated as a quadratic minimization problem:

$$\hat{\mathbf{z}}_0^{(t)}(\lambda_{\text{curr}}) = \arg\min_{\mathbf{z}} \ \lambda_{\text{curr}}\|\mathbf{z} - \hat{\mathbf{z}}_{\mathbf{y}}\|_2^2 + \lambda_{\text{ref}}\|\mathbf{z} - \hat{\mathbf{z}}_0^{(t)}\|_2^2, \tag{25}$$

which has a closed-form solution:

$$\mathbf{z} = \frac{\lambda_{\text{curr}}}{\lambda_{\text{curr}} + \lambda_{\text{ref}}}\hat{\mathbf{z}}_{\mathbf{y}} + \frac{\lambda_{\text{ref}}}{\lambda_{\text{curr}} + \lambda_{\text{ref}}}\hat{\mathbf{z}}_0^{(t)}. \tag{26}$$

The weights $\lambda_{\text{curr}}$ and $\lambda_{\text{ref}}$ determine the relative trust between the optimized and reference-informed latents. To maintain consistency with diffusion reverse trajectories, these weights should adapt across timesteps. At early timesteps (near $T$=999), $\hat{\mathbf{z}}_0^{(t)}$ remains far from the true latent $\mathbf{z}_{\mathbf{y}}$, so excessive reliance on $\hat{\mathbf{z}}_{\mathbf{y}}$ would amplify semantic mismatches between masked and measurement regions, destabilizing the trajectory. Conversely, at later timesteps, $\hat{\mathbf{z}}_0^{(t)}$ converges closer to $\mathbf{z}_{\mathbf{y}}$, warranting increased weight on $\hat{\mathbf{z}}_{\mathbf{y}}$. This motivates a time-dependent weighting scheme where $\lambda_{\text{curr}}$ increases and $\lambda_{\text{ref}}$ decreases as denoising progresses.

Table 7 compares different weighting strategies. Using only $\hat{\mathbf{z}}_{\mathbf{y}}$ ($\lambda_{\text{curr}}$=1, $\lambda_{\text{ref}}$=0) severely degrades all metrics, confirming that measurement satisfaction alone is insufficient. Using only $\hat{\mathbf{z}}_0^{(t)}$ ($\lambda_{\text{curr}}$=0, $\lambda_{\text{ref}}$=1) improves unpaired metrics but sacrifices paired performance. Fixed symmetric weighting ($\lambda_{\text{curr}}$=$\lambda_{\text{ref}}$) provides balanced results but does not account for trajectory evolution. Time-dependent schemes align better with diffusion dynamics: $\lambda_{\text{curr}} = \frac{\lambda_{\text{ref}}\bar{\alpha}_{t-1}}{1 - \bar{\alpha}_{t-1}}$ (our default, highlighted in gray) achieves the best unpaired metrics (FID, KID) and competitive paired metrics, while

| ④ Data consist. | | SSIM ↑ | PSNR ↑ | LPIPS ↓ | FID ↓ | KID ↓ |
|---|---|---|---|---|---|---|
| $\lambda_{\text{curr}}$ | $\lambda_{\text{ref}}$ | | | | | |
| 1 | 0 | 0.8677 | 22.5991 | 0.1623 | 20.1817 | 0.0088 |
| 0 | 1 | 0.8552 | 23.1475 | 0.0833 | 10.4362 | 0.0014 |
| $\lambda_{\text{ref}}$ | $\lambda_{\text{curr}}$ | 0.8862 | 23.7223 | 0.0748 | 9.8565 | 0.0010 |
| $\frac{\lambda_{\text{ref}}(1-\bar{\alpha}_{t-1})}{\bar{\alpha}_{t-1}}$ | – | **0.8864** | **23.7327** | 0.0749 | 9.8938 | 0.0012 |
| $\frac{\lambda_{\text{ref}}\bar{\alpha}_{t-1}}{1-\bar{\alpha}_{t-1}}$ | – | 0.8859 | 23.7027 | **0.0746** | **9.7669** | **0.0009** |

Table 7: Ablation study on standard denoising frequency. Impact of periodic standard denoising interval $N$ (component (C) in Fig. 2). $N=2$ (gray row, our default) balances trajectory stability and measurement preservation. **Bold**: best, underline: second-best.

| (C) Std. denoising | SSIM ↑ | PSNR ↑ | LPIPS ↓ | FID ↓ | KID ↓ |
|---|---|---|---|---|---|
| $N = 1$ | **0.8860** | **23.7041** | 0.0748 | 9.8309 | 0.0013 |
| $N = 2$ | 0.8859 | 23.7027 | **0.0746** | **9.7669** | **0.0009** |
| $N = 3$ | 0.8858 | 23.6891 | **0.0746** | 9.7841 | 0.0014 |
| $N = 5$ | 0.8857 | 23.6780 | **0.0746** | 9.7687 | 0.0013 |
| $N = 10$ | 0.8855 | 23.6602 | **0.0746** | 9.8106 | 0.0016 |
| $N = 25$ | 0.8852 | 23.6387 | 0.0747 | 9.9489 | 0.0012 |
| $N = 50$ | 0.8552 | 23.1475 | 0.0833 | 10.4362 | 0.0014 |

Table 8: Ablation study on data consistency interpolation weights. Comparison of different weighting strategies for balancing $\hat{\mathbf{z}}_{\mathbf{y}}$ and $\hat{\mathbf{z}}_0^{(t)}$ in Eq. 6. Time-dependent weighting $\lambda_{\text{curr}} = \frac{\lambda_{\text{ref}}\bar{\alpha}_{t-1}}{1-\bar{\alpha}_{t-1}}$ (gray row, our default) achieves the best balance across metrics. **Bold**: best, underline: second-best.

$\lambda_{\text{curr}} = \frac{\lambda_{\text{ref}}(1-\bar{\alpha}_{t-1})}{\bar{\alpha}_{t-1}}$ performs best on paired metrics. We select the former as it better balances all metrics and produces more stable trajectories.

**Frequency of standard denoising.** Table 8 examines the frequency $N$ of periodic standard denoising steps (component (C)), which realigns trajectories with noisy data manifolds $\mathcal{M}_t$. Too frequent application ($N=1$) over-smooths measurement constraints, slightly degrading unpaired metrics. Too infrequent application ($N\geq25$) prevents sufficient optimization, degrading all metrics. $N=2$ (our default, highlighted in gray) provides the best balance, achieving optimal unpaired performance (FID, KID) while maintaining strong paired metrics. This confirms that moderate realignment frequency effectively stabilizes trajectories without over-smoothing measurement constraints.

## A.5 ADDITIONAL RESULTS

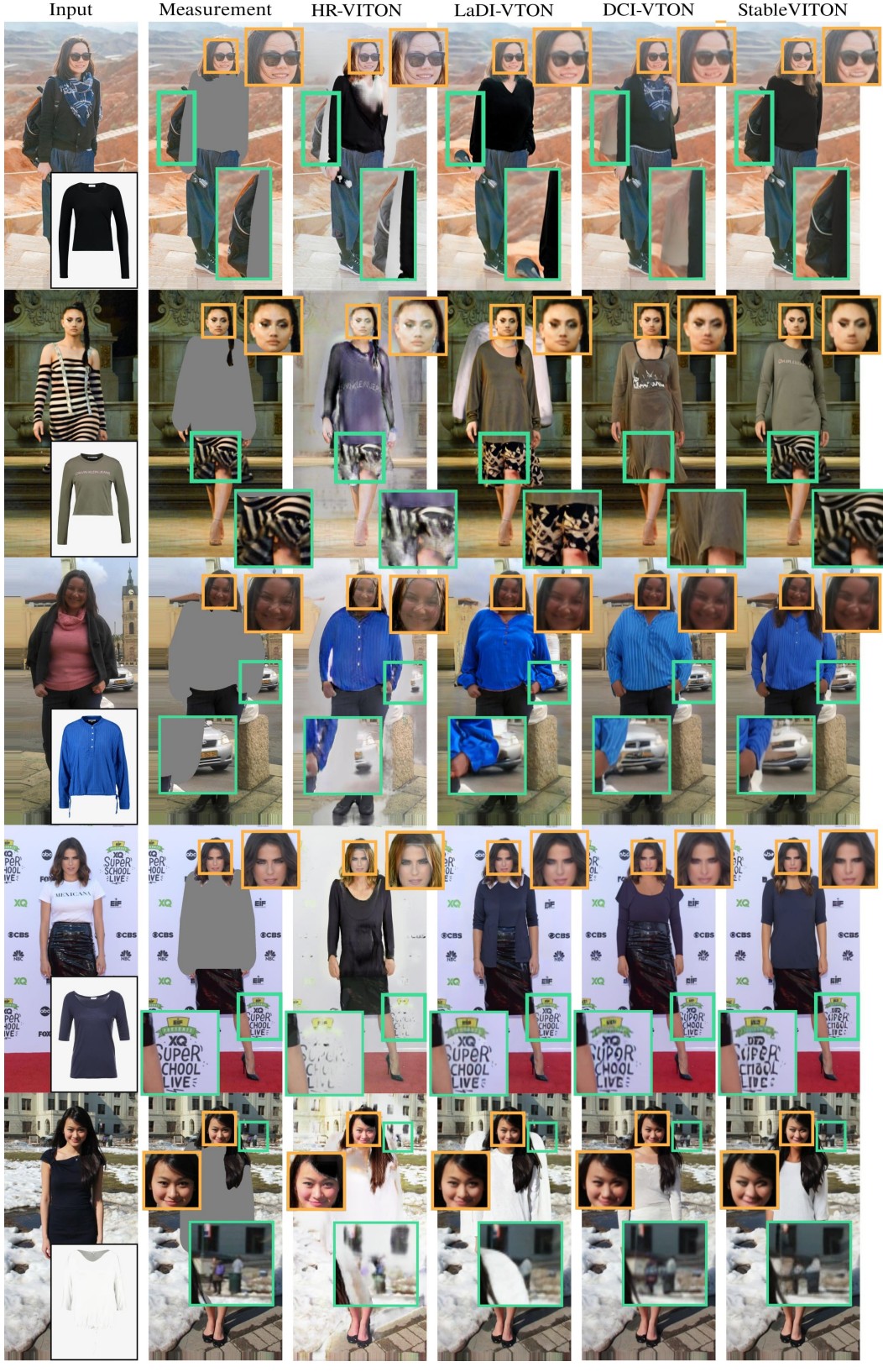

Figure 7: Qualitative results of baseline models on the SHHQ-1.0 dataset. Our observations show that generated images fail to preserve content in non-try-on regions: bags, skirts, cars, text, and human features (green boxes). Orange boxes indicate areas where facial details are not properly preserved.

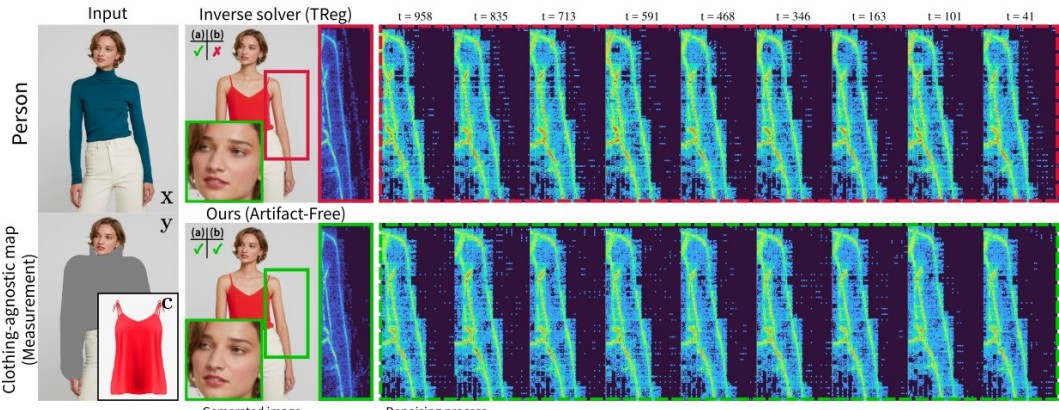

Figure 8: Extended comparison of artifact maps across timesteps during generation for the inverse solver (TReg) versus ART-VITON. We highlight semantic drift: TReg produces predominantly red maps, indicating persistent artifacts, while ART-VITON mitigates drift and satisfies both (a) artifact-free outputs and (b) measurement adherence, yielding mostly green maps. Solid and dashed boxes denote final and intermediate outputs, respectively.

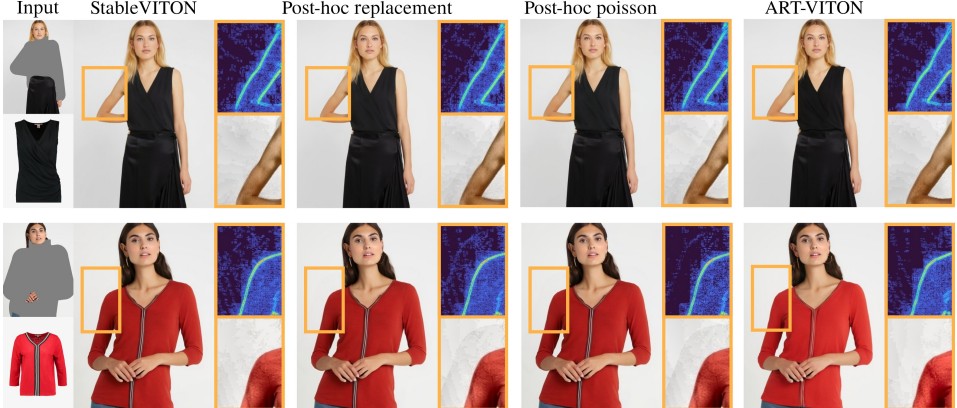

Figure 9: Comparison of post-hoc replacement, Poisson blending, and ART-VITON. Both post-hoc replacement and Poisson blending produce visible inconsistencies at region junctions (orange boxes: contrast-enhanced close-ups with artifact maps). Poisson blending smooths gradients in pixel space but often amplifies distortions by masking rather than resolving latent-space misalignment. ART-VITON substantially mitigates artifacts by addressing their root cause during diffusion sampling, producing visually coherent results with preserved fine details.

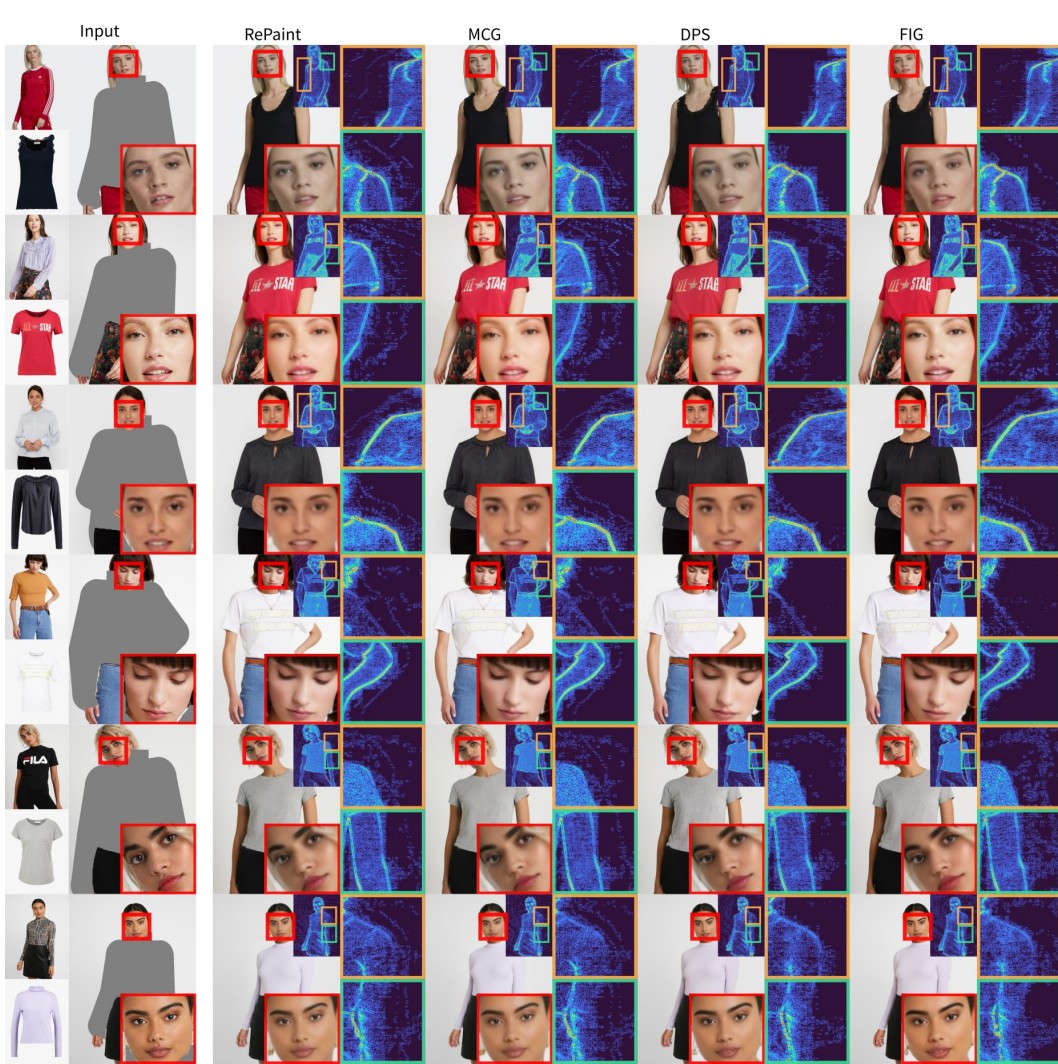

Figure 10: StableVITON on VITON-HD with inverse solvers applied without post-hoc replacement. Red indicates face zoom-in, and orange and green indicate artifact map zoom-ins. Hard constraint solvers (RePaint, MCG) and progressive update solvers (DPS, FIG) fail to fully satisfy measurements, highlighting the need for post-hoc replacement. Hard constraints generate artifacts due to semantic inconsistencies across regions, whereas progressive updates produce minimal artifacts, as each update induces only small changes.

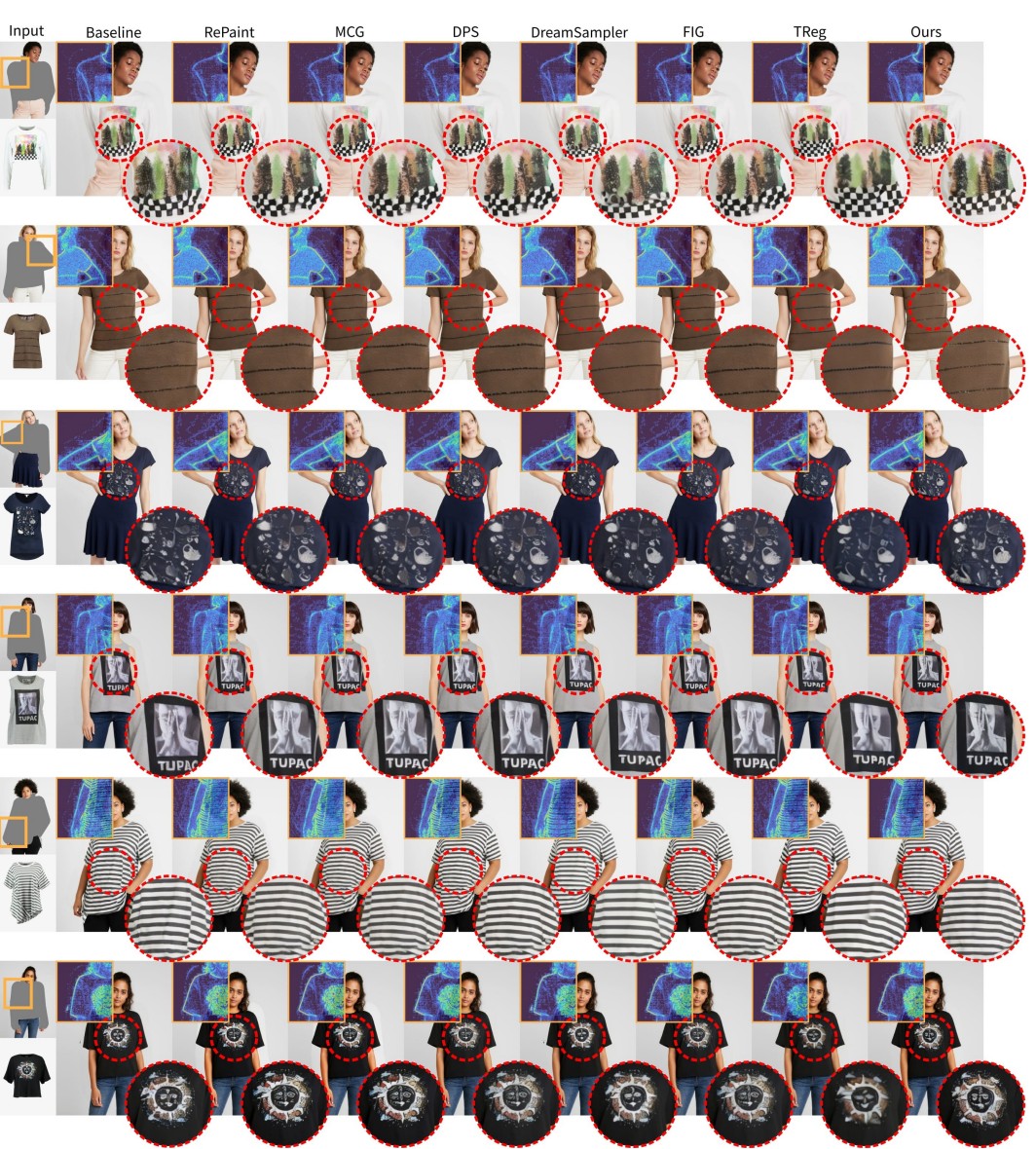

Figure 11: Comparison on the VITON-HD dataset with baseline (StableVITON) and existing inverse solvers. Red circles highlight texture degradation, particularly in hybrid stochastic methods (DreamSampler, TReg), while our approach preserves fine garment details and patterns. Orange boxes indicate artifacts present in other methods, which are absent in our results.

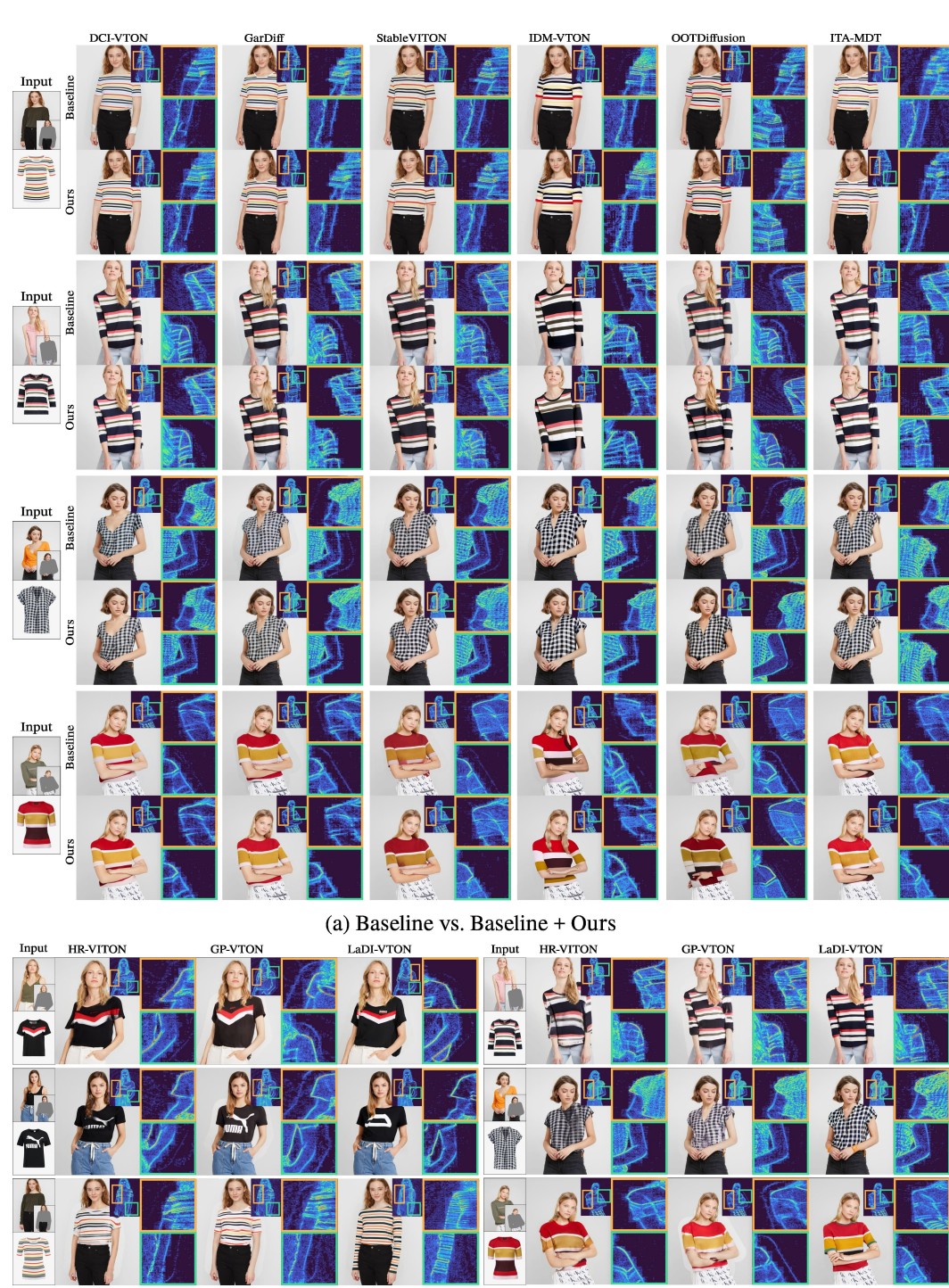

Figure 12: Additional qualitative results on the VITON-HD comparing baseline methods with our approach. (a) Comparison of baselines and their versions enhanced with our method: our approach consistently removes boundary artifacts while preserving high-frequency garment details such as logos, text, and complex patterns. (b) Results of the remaining models without our enhancement: in 2-stage pipeline models, warping results show garment distortions and color inconsistencies.

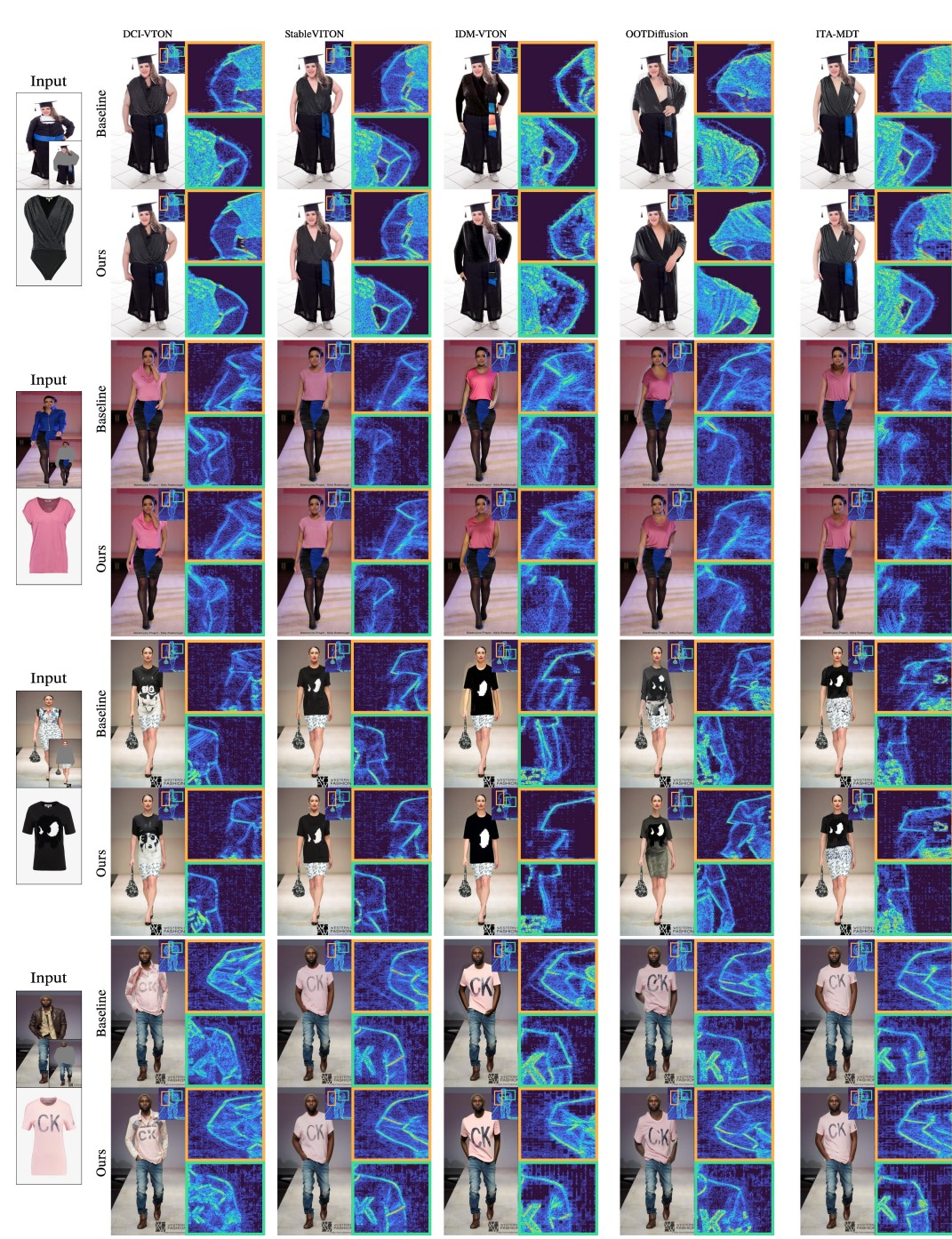

Figure 13: Extended comparison demonstrating robustness across domains on the SHHQ-1.0 dataset. (a) Comparison of baselines and their versions enhanced with our method: even in cross-domain scenarios, our approach effectively removes artifacts, demonstrating robustness. (b) Other VITON methods show boundary artifacts and garment distortion, whereas our approach preserves boundaries and garment details.

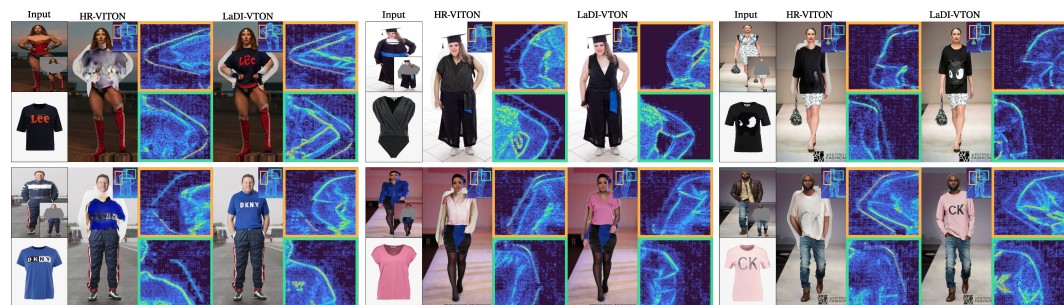

Figure 14: Extended comparison demonstrating robustness across domains on the SHHQ-1.0. Other VITON methods show boundary artifacts and garment distortion.

Figure 15: Qualitative comparison of baseline VITON methods on DressCode. Traditional methods (GP-VTON, LaDI-VTON) exhibit misalignment and texture distortion, while StableVITON shows boundary artifacts despite better garment alignment. Our method applied to StableVITON (right-most) alleviates boundary inconsistencies while preserving garment details and identity features.

