# OpenReview forum: "ART-VITON: Measurement-Guided Latent Diffusion for Artifact-Free Virtual Try-On"
_ICLR.cc/2026/Conference — ICLR 2026 Conference Withdrawn Submission_

### Official Review · Reviewer_JK6a · 2025-10-24

**Soundness:** 2
**Presentation:** 2
**Contribution:** 1
**Rating:** 2
**Confidence:** 5

**Summary:**

This paper proposes ART-VITON, a measurement-guided latent diffusion framework for artifact-free virtual try-on. The central idea is to recast virtual try-on as a linear inverse problem and introduce a latent diffusion-based inverse solver that incorporates measurement adherence progressively during the sampling process. ART-VITON combines residual prior-based initialization, artifact-free measurement-guided sampling utilizing data consistency and frequency-level correction, and periodic standard denoising.

**Strengths:**

1. Clear Motivation: The paper addresses a well-recognized, often neglected issue in virtual try-on—preserving non-try-on regions and avoiding boundary artifacts between try-on and non-try-on areas.
2 Sufficient Ablation Study: Ablation studies (Table 5 & 6, Figure 5) provide convincing evidence for their necessity and complementary effects.

**Weaknesses:**

1. The baseline methods does not cover the latest approaches like IDM-VTON[1] or OOTDiffusion[2].
[1] Improving diffusion models for authentic virtual try-on in the wild.
[2] Ootdiffusion: Outfitting fusion based latent diffusion for controllable virtual try-on.

2. Limited theoretical novelty in inverse solvers.  Although reducing abrupt changes innon-try-on regions is useful in Try-on, the core of the proposed high-frequency correction is to restrict the editing region with the mask, which is a common technique. It would be much better if the idea of partial editing in LDM is validated in more complex tasks.

3. The overall quality is not good enough for ICLR.

**Questions:**

See weakness.

---

> ### Author Response · Authors · 2025-11-22
> **Response to Reviewer JK6a (1/3)**
>
> **[W1] The baseline methods does not cover the latest approaches like IDM-VTON[1] or OOTDiffusion[2]. [1] Improving diffusion models for authentic virtual try-on in the wild.**
>
> > - We additionally conducted experiments on two models mentioned by the reviewer—**IDM-VTON [1] and OOTDiffusion [2]**—as well as a recently proposed VITON model, **ITA-MDT [3]**. These models, along with our original baselines, cover a broad range of latent diffusion frameworks:
> >    - Stable Diffusion v1.4 [7]/v2.1: DCI-VTON [4], GarDiff [5], StableVITON [6]
> >    - SD v1.5: OOTDiffusion
> >    - SDXL [8]: IDM-VTON
> >    - Masked Diffusion Transformer [9]: ITA-MDT
> > - We selected OOTDiffusion instead of IMAGDressing-v1 [10], as both rely on SD v1.5 and therefore provide equivalent architectural coverage.
> > - Implementation details are provided in **Appendix 2: Implementation Details of Baselines.**
> > - Our solver consistently improves all models across quantitative metrics **(Table 3)** and qualitatively mitigates artifacts across diverse architectures **(Figures 4, 5, 12-15)**, validating that ART-VITON is model-agnostic and plug-and-play compatible with transformer-based and SDXL pipelines as well. We note, however, that while improvements are substantial, small residual artifacts remain in some difficult cases (e.g., OOTDiffusion in challenging scenarios, **Figure 5**).
> > - Accordingly, we have updated **Section 5.3 (Comparison with VITON Baselines)** to more accurately describe our result, replacing terms such as "artifact-free" with "substantially mitigates artifacts" or "near-artifact-free in most cases". We explicitly acknowledge this **limitation** in the Conclusion, clarifying that fully eliminating artifacts across extreme cases remains an open challenge and promising future research direction.
> > - Our core findings remain valid: While existing methods (post-hoc replacement, inverse solvers) either fail to sufficiently reduce artifacts or introduce new distortions, our approach consistently mitigates artifacts across all tested baselines—including three recent models (IDM-VTON, OOTDiffusion, ITA-MDT)—while maintaining semantic coherence and avoiding performance trade-offs.
> > - Importantly, we observe that improvement correlates with baseline quality: models with moderate artifacts achieve near-complete artifact mitigation, while those with severe artifacts show substantial but incomplete improvement. We explicitly acknowledge this **limitation** in the **Conclusion**, emphasizing that complete artifact elimination across all conditions remains a challenging problem.
> >    > Limitations. *While ART-VITON effectively reduces boundary artifacts, its performance depends on the quality of the underlying baseline. Models with severe inherent artifacts—such as major semantic drift or measurement violations—benefit from our method but may retain minor imperfections. Combining our solver with stronger baselines could further improve results. Nevertheless, the model-agnostic design ensures consistent gains across all baselines.*
>
> [1] Choi, Yisol, et al. "Improving diffusion models for authentic virtual try-on in the wild." *European Conference on Computer Vision*. Cham: Springer Nature Switzerland, 2024.
>
> [2] Xu, Yuhao, et al. "Ootdiffusion: Outfitting fusion based latent diffusion for controllable virtual try-on." *Proceedings of the AAAI Conference on Artificial Intelligence*. Vol. 39. No. 9. 2025.
>
> [3] Hong, Ji Woo, et al. "ITA-MDT: Image-Timestep-Adaptive Masked Diffusion Transformer Framework for Image-Based Virtual Try-On." *Proceedings of the Computer Vision and Pattern Recognition Conference*. 2025.
>
> [4] Gou, Junhong, et al. "Taming the power of diffusion models for high-quality virtual try-on with appearance flow." Proceedings of the 31st ACM International Conference on Multimedia. 2023.
>
> [5] Wan, Siqi, et al. "Improving virtual try-on with garment-focused diffusion models." *European Conference on Computer Vision*. Cham: Springer Nature Switzerland, 2024.
>
> [6] Kim, Jeongho, et al. "Stableviton: Learning semantic correspondence with latent diffusion model for virtual try-on." *Proceedings of the IEEE/CVF conference on computer vision and pattern recognition*. 2024.
>
> [7] Rombach, Robin, et al. "High-resolution image synthesis with latent diffusion models." *Proceedings of the IEEE/CVF conference on computer vision and pattern recognition*. 2022.
>
> [8] Podell, Dustin, et al. "Sdxl: Improving latent diffusion models for high-resolution image synthesis." *arXiv preprint arXiv:2307.01952* (2023).
>
> [9] Gao, Shanghua, et al. "Mdtv2: Masked diffusion transformer is a strong image synthesizer." *arXiv preprint arXiv:2303.14389* (2023).
>
> [10] IMAGDressing-v1: *Customizable Virtual Dressing*, arXiv:2407.12705

---

> > ### Comment · Reviewer_JK6a · 2025-11-26
> >
> > Thanks for authos' response and great effort for the additional experiments. The motivation of the paper is clear and the proposed does avoid boundary artifacts. However, the contribution of this paper is not enough. (1) The application remains only in the try-on task. It is recommended to validate the proposed method  on the general inpainting tasks. (2) Mask-based process is not necessary for the try-on task with the strong foundation models.  Foe example, AnyDressing customizes character images based on arbitrary clothing combinations and personalized text prompts.
> >
> > Considering the above reasons, I will maintain the score.

---

> > > ### Author Response · Authors · 2025-12-01
> > > **To Reviewer Jk6a**
> > >
> > > - We sincerely appreciate the reviewer’s careful evaluation and constructive feedback. We understand the concerns regarding the scope and theoretical contribution. Still, we would like to emphasize that our method directly addresses a practical yet under-explored limitation of diffusion-based try-on systems—high-frequency inconsistencies and boundary artifacts—which existing foundation models and prior solvers continue to struggle with. Our approach remains lightweight, plug-and-play, and compatible with current and future foundation models, and the additional experiments further validate its effectiveness.
> > > - We also acknowledge the reviewer’s suggestions on extending validation to general inpainting tasks and evaluating models that do not rely on mask-based processing. These are valuable directions and align with our future research plans. We note, however, that the boundary artifacts we target fundamentally arise when non-try-on regions are not accurately generated. Therefore, even powerful models like AnyDressing [1] may still face difficulties in faithfully synthesizing these regions, and similar issues may occur, although we did not conduct experiments for this scenario.
> > > - We thank the reviewer again for their time and thoughtful assessment.
> > >
> > > [1] Li, Xinghui, et al. "Anydressing: Customizable multi-garment virtual dressing via latent diffusion models." 2025 IEEE/CVF Conference on Computer Vision and Pattern Recognition (CVPR). IEEE, 2025.

---

> ### Author Response · Authors · 2025-11-22
> **Response to Reviewer JK6a (2/3)**
>
> **[W2] Limited theoretical novelty in inverse solvers. Although reducing abrupt changes innon-try-on regions is useful in Try-on, the core of the proposed high-frequency correction is to restrict the editing region with the mask, which is a common technique. It would be much better if the idea of partial editing in LDM is validated in more complex tasks.**
>
> > - Thank you for this thoughtful comment. We believe there is a misunderstanding regarding the role of masking in our high-frequency correction module, and we clarify the contribution as follows.
> > - First, masking is not the core mechanism of our high-frequency correction.
> >    - The reviewer’s comment suggests that the high-frequency correction mainly relies on restricting the edited region via a mask, which is a common technique. However, this is not the essence of our method.
> >    - The fundamental challenge we address originates from the high-frequency degradation caused by VAE re-encoding during the construction of the optimized latent $\\hat{\\mathbf{z}}\_{\\mathbf{y}}$ in **(Eq. 6)**:
> >    $$\\min\_{\\mathbf{z}} \\left\\| \\frac{\\hat{\\mathbf{z}}\_0^{(t)} - \\mathcal{E}(\\mathcal{D}(\\mathbf{z}))}{2\\sigma\_{\\mathcal{E}}^2} \\right\\|\_2^2, \\hat{\\mathbf{z}}\_0^{(t)}(\\bar \\alpha\_{t-1}) = \\bar \\alpha\_{t-1} \\hat{\\mathbf{z}}\_{\\mathbf{y}} + (1 - \\bar \\alpha\_{t-1})\\hat {\\mathbf{z}}\_0^{(t)} -(6)$$
> > - Because $\\hat{\\mathbf{z}}\_{\\mathbf{y}}$ is generated via VAE encoding-decoding, it suffers from loss of high-freq components, a widely reported limitation in diffusion-based image synthesis. When used in **Eq. (6)**, this degraded latent propagates blurring and texture loss **across the entire latent field**, including masked (try-on) regions. Thus, *relying solely on restricting edits via a mask*, as implied in the reviewer’s comment, cannot resolve this issue, since the high-frequency degradation already affects both regions globally.
> > - Our method restores high-freq information by fusing the low-freq structure of $\\hat{\\mathbf{z}}\_{\\mathbf{y}}$ with the high-freq components from the reference-informed latent $\\hat{\\mathbf{z}}\_{\\mathbf{0}}^{(t)}$ in the frequency domain across both measurement and masked regions. This yields:
> >     1. Measurement region: recovers missing high-frequency details lost by VAE compression.
> >     2. Masked region: preserves its intact high-frequency content via selective replacement.
> >     3. Overall cross-region consistency: ensures semantic alignment across regions, preventing the mismatched sharpness that leads to artifacts.
> > - Therefore, the novelty lies in **global high-frequency restoration across all regions without retraining**, rather than region restriction, enabling coherent synthesis within the inverse-solver formulation.
> > - Regarding “validation on more complex tasks”:
> >     - We fully appreciate the reviewer’s suggestion. However, we would like to emphasize that VITON is itlsef a highly demanding partial-editing task, requiring
> >         1. Strict measurement preservation (identity, pose, background—regions that must remain unchanged),
> >         2. Precise geometric and photometric garment synthesis, and
> >         3. Seamless semantic alignment between preserved and edited regions, where even small mismatches produce visible seams.
> >     - To validate generality within the VITON task, we expanded experiments to include three more state-of-the-art architectures —IDM-VTON [1], OOTDiffusion [2], and ITA-MDT [3]—covering diverse architectures (SD [4] v1.4, v1.5, v2.1, SDXL [5], and Masked Diffusion Transformer [6]). As shown in **Table 3** and **Figures 4, 5, 12-15**, our method consistent improves all baselines without trade-offs, demonstrating robustness across diverse architectures. We acknowledge that applying this framework to other editing tasks is a promising direction and have added this point to the **Conclusion** as future work.
>
> [1] Choi, Yisol, et al. "Improving diffusion models for authentic virtual try-on in the wild." *European Conference on Computer Vision*. Cham: Springer Nature Switzerland, 2024.
>
> [2] Xu, Yuhao, et al. "Ootdiffusion: Outfitting fusion based latent diffusion for controllable virtual try-on." *Proceedings of the AAAI Conference on Artificial Intelligence*. Vol. 39. No. 9. 2025.
>
> [3] Hong, Ji Woo, et al. "ITA-MDT: Image-Timestep-Adaptive Masked Diffusion Transformer Framework for Image-Based Virtual Try-On." *Proceedings of the Computer Vision and Pattern Recognition Conference*. 2025.
>
> [4] Rombach, Robin, et al. "High-resolution image synthesis with latent diffusion models." *Proceedings of the IEEE/CVF conference on computer vision and pattern recognition*. 2022.
>
> [5] Podell, Dustin, et al. "Sdxl: Improving latent diffusion models for high-resolution image synthesis." *arXiv preprint arXiv:2307.01952* (2023).
>
> [6] Gao, Shanghua, et al. "Mdtv2: Masked diffusion transformer is a strong image synthesizer." *arXiv preprint arXiv:2303.14389* (2023).

---

> ### Author Response · Authors · 2025-11-22
> **Response to Reviewer JK6a (3/3)**
>
> **[W3] The overall quality is not good enough for ICLR.**
>
> > - We sincerely thank the reviewer for the feedback. We respectfully request, however, more specific references to sections, figures, or methodological weaknesses. We would very much appreciate more actionable feedback to further improve our work.
> > - We have carefully addressed the detailed comments provided by the other reviewers, which significantly strengthened the technical clarity, experimental rigor, and overall presentation quality of the paper.
> > - Methodological Clarity and Completeness:
> >     - **Expanded ablation** for key design choices, including interpolation weights in **Eq. 6 (Table 7)** and the frequency of periodic denoising **(Table 8)**, along with theoretical justifications for our design choices.
> >    $$\\min\_{\\mathbf{z}} \\left\\| \\frac{\\hat{\\mathbf{z}}\_0^{(t)} - \\mathcal{E}(\\mathcal{D}(\\mathbf{z}))}{2\\sigma\_{\\mathcal{E}}^2} \\right\\|\_2^2, \\hat{\\mathbf{z}}\_0^{(t)}(\\bar \\alpha\_{t-1}) = \\bar \\alpha\_{t-1} \\hat{\\mathbf{z}}\_{\\mathbf{y}} + (1 - \\bar \\alpha\_{t-1})\\hat {\\mathbf{z}}\_0^{(t)} -(6)$$
> >    - **Motivation for prior-based Initialization**, explicitly connecting it to the inverse problem formulation in Eq. 4 and demonstrating its role in stable posterior estimation.
> >    $$p(\\mathbf{x} | \\mathbf{y}, \\hat{\\mathbf{z}}\_0^{(t)}) \\propto p(\\hat{\\mathbf{z}}\_0^{(t)} | \\mathcal{D}(\\mathbf{z}), \\mathbf{y}) \\cdot p(\\mathbf{y} | \\mathcal{D}(\\mathbf{z})) - (4)$$
> > - Implementation details and Experimental rigor: To ensure reproducibility, we expanded the **Appendix 2: Implementation details of baselines** to include full hyper-parameters, baseline configurations.
> >     - **Extended Baselines**: We added three recently proposed models (IDM-VTON [1] , OOTDiffusion [2], ITA-MDT [3]) to demonstrate the model-agnostic effectiveness of our method across diverse architectures (SD [4] v1.4/v1.5/v2.1, SDXL [5], Masked Diffusion Transformer [6]).
> >     - **Post-hoc Comparison**: We performed a systematic comparison with Poisson Blending **(Table 3, Figure 9)** to illustrate why gradient-domain approaches fail in the VITON setting.
> >     - **Computational Efficiency**: We added TFLOPs, inference time and memory usage analyses **(Table 2)**, showing that our method maintains reasonable computational cost and comparable inference speed to existing solvers (~12s) while achieving superior quality.
> > - Presentation improvements
> >     - **Revised Overclaims**: refined language to avoid overstated claims (e.g.,, replacing “artifact-free” with “substantially mitigates artifacts”), and explicitly stated limitations in the Conclusion.
> >     - **Figure Clarity**: We added mask visualizations to **Figure 2**, clarified caption structures, and provided detailed explanations of the components.
> > - We hope these revisions sufficiently demonstrate our effort to substantially improve clarity and completeness. If the reviewer could kindly indicate specific areas where further enhancements would be most beneficial, we would be pleased to address them.
>
> [1] Choi, Yisol, et al. "Improving diffusion models for authentic virtual try-on in the wild." *European Conference on Computer Vision*. Cham: Springer Nature Switzerland, 2024.
>
> [2] Xu, Yuhao, et al. "Ootdiffusion: Outfitting fusion based latent diffusion for controllable virtual try-on." *Proceedings of the AAAI Conference on Artificial Intelligence*. Vol. 39. No. 9. 2025.
>
> [3] Hong, Ji Woo, et al. "ITA-MDT: Image-Timestep-Adaptive Masked Diffusion Transformer Framework for Image-Based Virtual Try-On." *Proceedings of the Computer Vision and Pattern Recognition Conference*. 2025.
>
> [4] Rombach, Robin, et al. "High-resolution image synthesis with latent diffusion models." *Proceedings of the IEEE/CVF conference on computer vision and pattern recognition*. 2022.
>
> [5] Podell, Dustin, et al. "Sdxl: Improving latent diffusion models for high-resolution image synthesis." *arXiv preprint arXiv:2307.01952* (2023).
>
> [6] Gao, Shanghua, et al. "Mdtv2: Masked diffusion transformer is a strong image synthesizer." *arXiv preprint arXiv:2303.14389* (2023).

---

### Official Review · Reviewer_CHdC · 2025-10-27

**Soundness:** 2
**Presentation:** 2
**Contribution:** 2
**Rating:** 4
**Confidence:** 5

**Summary:**

This paper addresses the challenge in Virtual Try-On (VITON) of preserving non-try-on regions and eliminating boundary artifacts in latent diffusion models by reformulating the problem as a linear inverse problem.

The authors propose ART-VITON, a measurement-guided diffusion framework that ensures measurement adherence using residual prior-based initialization and an artifact-free sampling strategy.

Experiments demonstrate that ART-VITON effectively preserves identity and background, eliminates boundary artifacts, and consistently improves visual fidelity and robustness over state-of-the-art baselines.

**Strengths:**

1. Reformulates the virtual try-on task as a **linear inverse problem**, offering a principled approach to handle measurement consistency and reduce boundary artifacts.
2. Introduces a **measurement-guided diffusion framework** with residual prior initialization and frequency-level correction to ensure artifact-free, identity-preserving synthesis.
3. Demonstrates consistent performance gains and visual improvements across multiple benchmarks (VITON-HD, DressCode, SHHQ-1.0) compared to state-of-the-art methods.

**Weaknesses:**

1. Several implementation details remain unclear. For instance, Line 64 mentions *“violates measurements (M)”*—it is not explained how **M** is defined or computed. The example showing *t = 835* also lacks justification for this choice, and results for other timesteps are not discussed. Moreover, training details and computational overhead compared to baselines are missing, making it difficult to assess reproducibility and efficiency.

2. The claim of producing *“artifact-free”* results seems overstated. In Figure 1, minor artifacts are still visible, even if reduced compared to baselines. Similarly, Figures 4(b), 8, 9, and 10 show that improvements over baselines are sometimes marginal, suggesting the enhancement is not consistently substantial.

3. The evaluation lacks comparisons with **recent diffusion transformer-based VITON methods**, such as ITA-MDT [1] and IMAGDressing-v1 [2], which would provide a fairer and more comprehensive validation of the proposed plug-and-play module.

[1] *ITA-MDT: Image-Timestep-Adaptive Masked Diffusion Transformer Framework for Image-Based Virtual Try-On*, CVPR 2025

[2] *IMAGDressing-v1: Customizable Virtual Dressing*, arXiv:2407.12705

**Questions:**

See weaknesses

---

> ### Author Response · Authors · 2025-11-22
> **Response to Reviewer CHdC (1/4)**
>
> **[W1] Several implementation details remain unclear. For instance, Line 64 mentions “violates measurements (M)”—it is not explained how M is defined or computed. The example showing t = 835 also lacks justification for this choice, and results for other timesteps are not discussed. Moreover, training details and computational overhead compared to baselines are missing, making it difficult to assess reproducibility and efficiency.**
>
> > - We thank the reviewer for these insightful comments. We have revised the manuscript to improve clarity and reproducibility, and we address each point below.
> > - **“Violates Measurements (M)” (L64) and Figure 1:**
> >    - In **Figure 1** and the accompanying text, the original phrase “**violates measurements (M)**” indicated whether the generated output preserves non-try-on regions or not. For example, the face region in StableVITON [1] outputs violates **M (red box)**, while post-hoc replacement or **ART-VITON** outputs satisfy **M (green box)**. To avoid confusion with the binary mask notation **$\\mathbf{M}$** introduced later in the **Method section**, we revised this evaluation notation in **Figure 1** to:
> >        - **(a)** Artifact-free outputs: boundary artifacts absent
> >        - **(b)** Measurement adherence: measurement regions preserved
> >    - We also updated the caption to reflect these criteria and indicate success/violation using green/red markers:
> >       > Caption: *“Comparison of boundary artifacts across methods. We evaluate two criteria: (a) artifact-free outputs and (b) adherence to measurements. StableVITON achieves (a) but fails in (b). Post-hoc replacement enforces (b) but introduces seams, breaking (a). Inverse solvers maintain (b) but suffer semantic drift, degrading (a) over time. ART-VITON satisfies both (a) and (b). Green: success(measurement adherence or artifact-free); red: violations or artifacts. Solid/Dashed boxes show final/intermediate (t=835) outputs.”*
> > - Justification of timestep $t=835$ and additional analysis:
> >    - Output at timestep $t=835$ in **Figure 3** was was selected as a mid-sampling example to qualitatively illustrate semantic drift in inverse solvers. To provide a more complete understanding, we now include results across nine intermediate timesteps $(t=958 \\rightarrow 41)$ in **Figure 8 (Appendix)**. These results show that semantic drift and boundary artifacts accumulate progressively in baseline inverse solver TReg [2] **(red dotted boxes),** whereas ART-VITON maintains consistent trajectories throughout the sampling process **(green dotted boxes)**.
> > - Training details:
> >    - As stated in **Section 4**, ART-VITON is a *plug-and-play inverse solver* that requires *no additional training*. The optimization of our method consists of four steps: Prior-based initialization **(Section 4.2)**, Hard measurement constraint **(Section 4.3)**, Data consistency **(Section 4.3)** and High-frequency correction **(Section 4.3)**.
> >    - We use publicly released models without modifying architectural components. Full implementation parameters, inverse solver settings (RePaint [3], MCG [4], DPS [5], FIG [6], DreamSampler [7], TReg [2]) and base VITON models (DCI-VTON [8], GarDiff [9], StableVITON [1], IDM-VTON [10], OOTDiffusion [11], ITA-MDT [12]) are now included in **Appendices 2–3**.
> > - Computational overhead:
> >    - We now provide inference-time and memory usage comparison in **Table 2**. Using an NVIDIA RTX A5000 GPU on VITON-HD:
> > | Method               | SSIM ↑    | PSNR ↑    | LPIPS ↓  | FID ↓     | KID ↓   | **Inf.** (s) ↓ | **Mem.** (GB) ↓ |
> > |----------------------|-----------|-----------|----------|-----------|---------|------------|-------------|
> > | StableVITON (baseline) | 0.8839   | 23.5965   | 0.0757   | 9.8694    | 0.0016  | 9.117      | 7.66        |
> > | RePaint              | 0.8856    | 23.6635   | 0.0752   | 10.0829   | 0.0018  | 9.241      | 7.66        |
> > | MCG                  | 0.8855    | 23.6641   | 0.0752   | 10.085    | 0.0015  | 13.518     | 10.38       |
> > | DPS                  | 0.8851    | 23.6390   | 0.0749   | 9.9425    | 0.0014  | 13.573     | 10.38       |
> > | DreamSampler         | 0.8904    | **23.8984** | 0.0771   | 10.5143   | 0.0018  | 25.463     | 7.66        |
> > | FIG                  | 0.8851    | 23.6390   | 0.0749   | 9.9427    | 0.0014  | 13.389     | 10.38       |
> > | TReg                 | **0.8909** | 23.8205   | 0.0844   | 11.7467   | 0.0024  | 11.382     | 7.66        |
> > | Ours                 | 0.8859    | 23.7027   | **0.0746** | **9.7669** | **0.0009** | 12.101     | 7.66        |

---

> ### Author Response · Authors · 2025-11-22
> **Response to Reviewer CHdC (2/4)**
>
> > - Key observations of ART-VITON:
> >    - Comparable runtime (12.1s) to TReg (11.4s): only +0.7s, +6.1% overhead.
> >    - 2x faster than DreamSampler (22.5s)
> >    - Similar or faster than gradient-based methods (MCG, DPS, FIG: ~13.5s)
> >    - Lower memory (7.66 GB) than gradient-based optimization methods (~10.38 GB)
> >    - Moderate overhead, but significantly better performance consistency across metrics
> > - Efficiency analysis:
> >    - VAE encoding/decoding **(Eq. 5)** is required by other inverse solvers including TReg and DreamSampler, not unique to ART-VITON.
> >    - Latent-space optimization **(Eq. 6**) is grdient-free, adding negligible computation, thus memory-efficient.
> >    - Fourier-based frequency fusion **(Eq. 7)** constitutes the majority of added computation, but only 0.7s per image, enabling strong detail restoration without high cost.
> >    $$\\hat{\\mathbf{x}}\_{\\mathbf{y}} = \\mathbf{M} \\odot \\mathbf{y} + (1 - \\mathbf{M}) \\odot \\mathcal{D}(\\mathbf{z}) -(5)$$
> >    $$\\hat{\\mathbf{z}}\_0^{(t)}(\\bar \\alpha\_{t-1}) = \\bar \\alpha\_{t-1} \\hat{\\mathbf{z}}\_{\\mathbf{y}} + (1 - \\bar \\alpha\_{t-1})\\hat{\\mathbf{z}}\_0^{(t)} -(6)$$
> >    $$\\hat{\\mathbf{z}}\_0^{(t)}(\\bar \\alpha\_{t-1}) = \\mathbf{M}\odot (\\bar\\alpha\_{t-1}\\hat{\\mathbf{z}}\_{\\mathbf{y}}' + (1-\\bar\\alpha\_{t-1})\\hat{\\mathbf{z}}\_0^{(t)}) + (1-\\mathbf{M})\\odot \\hat{\\mathbf{z}}\_0^{(t)} - (7)$$
> > - While slightly slower than TReg due to Fourier fusion, ART-VITON maintains performance without trade-offs.
> > - Performance-Efficiency Balance:
> >    - Unlike existing methods that improve some aspects at the cost of others:
> >        - DreamSampler improves structural metrics (SSIM/PSNR) but severely harms perceptual similarity and FID/KID, while being slow (25.5s).
> >        - DPS/FIG improve FID but degrade SSIM/PSNR with runtime (~13.5s) and introduce boundary distortions.
> >        - ART-VITON consistently improves all metrics across paired and unpaired settings at moderate inference time (12.1s).
> > - Although ART-VITON performs additional operations per sampling step, the measured overhead is small and comparable to popular inverse solvers, while providing significantly more stable results without metric trade-offs.

---

> ### Author Response · Authors · 2025-11-22
> **Reference for W1.**
>
> [1] Kim, Jeongho, et al. "Stableviton: Learning semantic correspondence with latent diffusion model for virtual try-on." *Proceedings of the IEEE/CVF conference on computer vision and pattern recognition*. 2024.
>
> [2] Kim, Jeongsol, et al. "Regularization by Texts for Latent Diffusion Inverse Solvers." *The Thirteenth International Conference on Learning Representations*, 2025
>
> [3] Lugmayr, Andreas, et al. "Repaint: Inpainting using denoising diffusion probabilistic models." *Proceedings of the IEEE/CVF conference on computer vision and pattern recognition*. 2022.
>
> [4] Chung, Hyungjin, et al. "Improving diffusion models for inverse problems using manifold constraints." *Advances in Neural Information Processing Systems* 35 (2022): 25683-25696.
>
> [5] Chung, Hyungjin, et al. "Diffusion posterior sampling for general noisy inverse problems." *arXiv preprint arXiv:2209.14687* (2022).
>
> [6] Yan, Yici, et al. "FIG: Flow with Interpolant Guidance for Linear Inverse Problems." *The Thirteenth International Conference on Learning Representations*.
>
> [7] Kim, Jeongsol, Geon Yeong Park, and Jong Chul Ye. "Dreamsampler: Unifying diffusion sampling and score distillation for image manipulation." *European Conference on Computer Vision*. Cham: Springer Nature Switzerland, 2024.
>
> [8] Gou, Junhong, et al. "Taming the power of diffusion models for high-quality virtual try-on with appearance flow." Proceedings of the 31st ACM International Conference on Multimedia. 2023.
>
> [9] Wan, Siqi, et al. "Improving virtual try-on with garment-focused diffusion models." *European Conference on Computer Vision*. Cham: Springer Nature Switzerland, 2024.
>
> [10] Choi, Yisol, et al. "Improving diffusion models for authentic virtual try-on in the wild." *European Conference on Computer Vision*. Cham: Springer Nature Switzerland, 2024.
>
> [11] Xu, Yuhao, et al. "Ootdiffusion: Outfitting fusion based latent diffusion for controllable virtual try-on." *Proceedings of the AAAI Conference on Artificial Intelligence*. Vol. 39. No. 9. 2025.
>
> [12] Hong, Ji Woo, et al. "ITA-MDT: Image-Timestep-Adaptive Masked Diffusion Transformer Framework for Image-Based Virtual Try-On." *Proceedings of the Computer Vision and Pattern Recognition Conference*. 2025.

---

> ### Author Response · Authors · 2025-11-22
> **Response to Reviewer CHdC (3/4)**
>
> **[W2] The claim of producing “artifact-free” results seems overstated. In Figure 1, minor artifacts are still visible, even if reduced compared to baselines. Similarly, Figures 4(b), 8, 9, and 10 show that improvements over baselines are sometimes marginal, suggesting the enhancement is not consistently substantial.**
>
> > - We agree that the previous wording may have overstated the extent of improvement. While ART-VITON consistently reduces boundary artifacts more effectively than prior approaches, residual artifacts may still appear in baseline models with severe initial artifacts or in challenging conditions (e.g., long garments in OOTDiffusion [1], **Figure 5**, extreme poses or strong occlusions). Nonetheless, as shown in **Table 3**, our method demonstrates consistent improvements across all evaluated metrics.
> > - To address this, we revised the manuscript to use clearer descriptions and wording such as:
> >     - In **Section 5.3**: Comparison with VITON Baselines, terms like “artifact-free” are replaced with “substantially mitigates artifacts” or “near-artifact-free in most cases”, and replaces terms like “eliminates” with “substantially reduces” or “effectively suppresses”.
> > - Although improvements vary by baseline severity, the core findings remain meaningful and consistent: prior methods (post-hoc replacement, inverse solvers) either fail to sufficiently reduce artifacts or introduce new distortions. In contrast, our approach consistently maintains semantic coherence while mitigating artifacts across all tested baselines, including three additional state-of-the-art VITON models: IDM-VTON [2], OOTDiffusion [1], and ITA-MDT [3].
> > - Finally, we explicitly acknowledge in the **Conclusion** section that complete removal of artifacts remains a challenging open problem, and our work should be interpreted as a significant step toward this goal rather than its final resolution.
> >    > Limitations. *While ART-VITON effectively reduces boundary artifacts, its performance depends on the quality of the underlying baseline. Models with severe inherent artifacts—such as major semantic drift or measurement violations—benefit from our method but may retain minor imperfections. Combining our solver with stronger baselines could further improve results. Nevertheless, the model-agnostic design ensures consistent gains across all baselines.*
>
> [1] Xu, Yuhao, et al. "Ootdiffusion: Outfitting fusion based latent diffusion for controllable virtual try-on." *Proceedings of the AAAI Conference on Artificial Intelligence*. Vol. 39. No. 9. 2025.
>
> [2] Choi, Yisol, et al. "Improving diffusion models for authentic virtual try-on in the wild." *European Conference on Computer Vision*. Cham: Springer Nature Switzerland, 2024.
>
> [3] Hong, Ji Woo, et al. "ITA-MDT: Image-Timestep-Adaptive Masked Diffusion Transformer Framework for Image-Based Virtual Try-On." *Proceedings of the Computer Vision and Pattern Recognition Conference*. 2025.

---

> ### Author Response · Authors · 2025-11-22
> **Response to Reviewer CHdC (4/4)**
>
> **[W3] The evaluation lacks comparisons with recent diffusion transformer-based VITON methods, such as ITA-MDT [1] and IMAGDressing-v1 [2], which would provide a fairer and more comprehensive validation of the proposed plug-and-play module.**
>
> > - We have conducted additional experiments on three recent architectures, including **ITA-MDT [1]** as requested, as well as **IDM-VTON [2] and OOTDiffusion [3]**. These models, along with our original baselines, cover a broad range of latent diffusion frameworks:
> >    - Stable Diffusion v1.4 [7]/v2.1: DCI-VTON [4], GarDiff [5], StableVITON [6]
> >    - SD v1.5: OOTDiffusion
> >    - SDXL [8]: IDM-VTON
> >    - Masked Diffusion Transformer [9]: ITA-MDT
> > - We selected OOTDiffusion instead of IMAGDressing-v1 [10], as both rely on SD v1.5 and therefore provide equivalent architectural coverage.
> > - Implementation details are provided in **Appendix 2: Implementation Details of Baselines.**
> > - Our solver consistently improves all models across quantitative metrics **(Table 3)** and qualitatively mitigates artifacts across diverse architectures **(Figures 4, 5, 12-15)**, validating that ART-VITON is model-agnostic and plug-and-play compatible with transformer-based and SDXL pipelines as well. We note, however, that while improvements are substantial, small residual artifacts remain in some difficult cases (e.g., OOTDiffusion in challenging scenarios, **Figure 5**).
> > - Accordingly, we have updated **Section 5.3 (Comparison with VITON Baselines)** to more accurately describe our result, replacing terms such as "artifact-free" with "substantially mitigates artifacts" or "near-artifact-free in most cases". We explicitly acknowledge this **limitation** in the Conclusion, clarifying that fully eliminating artifacts across extreme cases remains an open challenge and promising future research direction.
> > - Our core findings remain valid: While existing methods (post-hoc replacement, inverse solvers) either fail to sufficiently reduce artifacts or introduce new distortions, our approach consistently mitigates artifacts across all tested baselines—including three recent models (IDM-VTON, OOTDiffusion, ITA-MDT)—while maintaining semantic coherence and avoiding performance trade-offs.
> > - Importantly, we observe that improvement correlates with baseline quality: models with moderate artifacts achieve near-complete artifact mitigation, while those with severe artifacts show substantial but incomplete improvement. We explicitly acknowledge this **limitation** in the **Conclusion**, emphasizing that complete artifact elimination across all conditions remains a challenging problem.
> >    > Limitations. *While ART-VITON effectively reduces boundary artifacts, its performance depends on the quality of the underlying baseline. Models with severe inherent artifacts—such as major semantic drift or measurement violations—benefit from our method but may retain minor imperfections. Combining our solver with stronger baselines could further improve results. Nevertheless, the model-agnostic design ensures consistent gains across all baselines.*
>
> [1] Hong, Ji Woo, et al. "ITA-MDT: Image-Timestep-Adaptive Masked Diffusion Transformer Framework for Image-Based Virtual Try-On." *Proceedings of the Computer Vision and Pattern Recognition Conference*. 2025.
>
> [2] Choi, Yisol, et al. "Improving diffusion models for authentic virtual try-on in the wild." *European Conference on Computer Vision*. Cham: Springer Nature Switzerland, 2024.
>
> [3] Xu, Yuhao, et al. "Ootdiffusion: Outfitting fusion based latent diffusion for controllable virtual try-on." *Proceedings of the AAAI Conference on Artificial Intelligence*. Vol. 39. No. 9. 2025.
>
> [4] Gou, Junhong, et al. "Taming the power of diffusion models for high-quality virtual try-on with appearance flow." Proceedings of the 31st ACM International Conference on Multimedia. 2023.
>
> [5] Wan, Siqi, et al. "Improving virtual try-on with garment-focused diffusion models." *European Conference on Computer Vision*. Cham: Springer Nature Switzerland, 2024.
>
> [6] Kim, Jeongho, et al. "Stableviton: Learning semantic correspondence with latent diffusion model for virtual try-on." *Proceedings of the IEEE/CVF conference on computer vision and pattern recognition*. 2024.
>
> [7] Rombach, Robin, et al. "High-resolution image synthesis with latent diffusion models." *Proceedings of the IEEE/CVF conference on computer vision and pattern recognition*. 2022.
>
> [8] Podell, Dustin, et al. "Sdxl: Improving latent diffusion models for high-resolution image synthesis." *arXiv preprint arXiv:2307.01952* (2023).
>
> [9] Gao, Shanghua, et al. "Mdtv2: Masked diffusion transformer is a strong image synthesizer." *arXiv preprint arXiv:2303.14389* (2023).
>
> [10] IMAGDressing-v1: *Customizable Virtual Dressing*, arXiv:2407.12705

---

> ### Author Response · Authors · 2025-11-25
> **[Updated Response to W1] Computational Cost Analysis with FLOPs**
>
> **[W1] Several implementation details remain unclear. For instance, Line 64 mentions “violates measurements (M)”—it is not explained how M is defined or computed. The example showing t = 835 also lacks justification for this choice, and results for other timesteps are not discussed. Moreover, training details and computational overhead compared to baselines are missing, making it difficult to assess reproducibility and efficiency.**
>
> > - We have added FLOPs (floating-point operations) measurements to **Table 2** to provide a complete computational efficiency analysis:
> > | Method | SSIM ↑ | PSNR ↑ | LPIPS ↓ | FID ↓ | KID ↓ | FLOPs (T) ↓ | Inf. (s) ↓ | Mem. (GB) ↓ |
> > |--------|--------|--------|---------|-------|-------|-------------|------------|--------------|
> > | StableVITON (baseline) | 0.8839 | 23.5965 | 0.0757 | 9.8694 | 0.0016 | 86.178 | 9.117 | 7.66 |
> > | RePaint | 0.8856 | 23.6635 | 0.0752 | 10.0829 | 0.0018 | 87.864 | 9.241 | 7.66 |
> > | MCG | 0.8855 | 23.6641 | 0.0752 | 10.085 | 0.0015 | 259.87 | 13.518 | 10.38 |
> > | DPS | 0.8851 | 23.6390 | 0.0749 | 9.9425 | 0.0014 | 259.87 | 13.573 | 10.38 |
> > | DreamSampler | 0.8904 | 23.8984 | 0.0771 | 10.5143 | 0.0018 | 267.22 | 25.463 | 7.66 |
> > | FIG | 0.8851 | 23.6390 | 0.0749 | 9.9427 | 0.0014 | 259.87 | 13.389 | 10.38 |
> > | TReg | 0.8909 | 23.8205 | 0.0844 | 11.7467 | 0.0024 | 130.63 | 11.382 | 7.66 |
> > | Ours | 0.8859 | 23.7027 | 0.0746 | 9.7669 | 0.0009 | 130.63 | 12.101 | 7.66 |
> > -  ART-VITON computational cost (130.63 TFLOPs):
> >    - Identical to TReg (130.63 TFLOPs)
> >    * 1.5× higher than baseline (86.178 TFLOPs)
> >    * 2× lower than gradient-based methods (MCG, DPS, FIG: ~260 TFLOPs)
> >    * 2× lower than DreamSampler (267.22 TFLOPs)
> > * Breakdown of additional FLOPs:
> >    * The primary overhead arises from the repeated VAE encoding/decoding steps.
> >    * Although inherently expensive, these iterations are essential for effective pixel optimization.
> > * Efficiency-Quality Trade-off:
> >    * Gradient-based methods require 2× FLOPs but still fail to improve unpaired metrics
> >    * DreamSampler has highest FLOPs (267.22 TFLOPs) yet degrades perceptual quality
> >    * ART-VITON achieves best overall performance with moderate FLOPs, demonstrating efficient use of computation
> > - The FLOPs measurements confirm that ART-VITON maintains reasonable computational cost while being the only method to consistently improve all metrics without trade-offs.

---

### Official Review · Reviewer_Jd4t · 2025-10-31

**Soundness:** 3
**Presentation:** 2
**Contribution:** 2
**Rating:** 4
**Confidence:** 4

**Summary:**

This paper proposes ART-VITON, a framework for virtual try-on aimed at resolving the boundary artifacts produced by existing methods when preserving non-try-on regions. The paper reformulates VITON as a linear inverse problem, seeking to reconstruct the full image conditioned on the given non-try-on regions (the "measurement"). The authors design an "artifact-free measurement-guided sampling" strategy that progressively guides the LDM's sampling trajectory during reverse diffusion to match these measurements.

**Strengths:**

The artifact-free measurement-guided solver combines multiple complementary techniques: (i) a residual prior-based initialization to mitigate the training-inference mismatch , (ii) data consistency to maintain semantic coherence , (iii) frequency-level correction to restore high-frequency details lost during VAE encoding , and (iv) periodic standard denoising to harmonize regions. Furthermore, the framework is model-agnostic, meaning it can be applied to various existing LDM-based VITON models without retraining, consistently improving their performance and demonstrating strong generalization on multiple datasets.

**Weaknesses:**

My primary concern stems from the paper's comparison against post-hoc replacement. The authors motivate their complex measurement-guided sampling framework (ART-VITON) by highlighting the "abrupt transitions" and "boundary artifacts" produced by standard "post-hoc replacement". However, the "post-hoc replacement" method described appears to be a naive pixel-wise copy-paste. To my knowledge, methods like CAT-DM[1] have effectively utilized Poisson Blending as a "post-hoc replacement"to resolve these exact artifacts. The authors should therefore discuss blending techniques, especially Poisson Blending, to situate their contribution more accurately.

Furthermore, the proposed sampling method seems extremely complex. Each sampling step involves: (1) a VAE decode and encode (Eq. 5), (2) a latent-space optimization (Eq. 6), and (3) both a Fourier Transform and an Inverse Fourier Transform (Eq. 7). This process is almost certainly much slower than standard DDIM sampling or even other solvers like TReg. The authors should include a discussion of the model's inference time and computational overhead.

[1] Zeng J, Song D, Nie W, et al. Cat-dm: Controllable accelerated virtual try-on with diffusion model[C]//Proceedings of the IEEE/CVF conference on computer vision and pattern recognition. 2024: 8372-8382.


Finally, Figure 2, as the main flowchart for the method, is overly dense and difficult to understand.

**Questions:**

Please address the issues raised in the weaknesses section. Including Poisson blending and inference time in the analysis would help address my concerns.

---

> ### Author Response · Authors · 2025-11-22
> **Response to Reviewer Jd4t (1/4)**
>
> **[W1] My primary concern stems from the paper's comparison against post-hoc replacement. The authors motivate their complex measurement-guided sampling framework (ART-VITON) by highlighting the "abrupt transitions" and "boundary artifacts" produced by standard "post-hoc replacement". However, the "post-hoc replacement" method described appears to be a naive pixel-wise copy-paste. To my knowledge, methods like CAT-DM[1] have effectively utilized Poisson Blending as a "post-hoc replacement"to resolve these exact artifacts. The authors should therefore discuss blending techniques, especially Poisson Blending, to situate their contribution more accurately.**
>
> > - Thank you for raising this important point. We agree that Poisson Blending is a meaningful reference point, and we have now included a comparison in **Table 3** and **Figure 9**, with additional discussion in **Section 5.3: Comparison with VITON Baselines (L460–L467)**. To assess whether gradient-domain blending can effectively resolve boundary artifacts in VITON, we applied Poisson Blending (used in CAT-DM[1]) to all VITON baselines.
> >    - **Table 3 (quantitative results)**: Poisson Blending yields inconsistent performance - while it sometimes increases paired structural metrics, it frequently degrades unpaired perceptual metrics (FID/KID) for most baselines (GarDiff [2], StableVITON [3], IDM-VTON [4], OOTDiffusion [5], ITA-MDT [6]). This suggests that gradient-domain smoothing disrupts semantic coherence established during generation, reducing overall perceptual quality.
> >    - **Figure 9 (qualitative results)**: Although Poisson Blending suppresses sharp pixel-level seams, it often produces texture mismatches and residual distortions at boundaries (contrast-enhanced close-ups in orange boxes). While post-hoc replacement exhibits noticeable boundary seams and color inconsistencies (orange boxes), **ART-VITON** maintains visual coherence across regions while preserving fine-grained details without boundary artifacts.
> >    - The last two columns correspond to SHHQ-1.0, while the remaining columns show evaluation results on the VITON-HD dataset. $*$ denotes post-hoc replacement, and $\\dagger$ denotes Poisson blending.
> >     | Method              | SSIM ↑ | PSNR ↑ | LPIPS ↓ | FID ↓   | KID ↓   | FID ↓   | KID ↓  |
> |-------------------|--------|--------|---------|---------|---------|---------|--------|
> | DCI-VTON*         | 0.8871 | 24.1413 | 0.0782  | 11.3634 | 0.0012  | 21.2350 | 0.0055 |
> | DCI-VTON†         | 0.8908 | 24.5180 | 0.0746  | 10.9724 | 0.0022  | 21.3168 | 0.0048 |
> | DCI-VTON (Ours)   | **0.8946** | **24.6903** | **0.0722** | 10.5408 | 0.0005  | **21.1485** | **0.0040** |
> | GarDiff*          | 0.8418 | 21.7263 | 0.0895  | 10.5858 | 0.0042  | -       | -       |
> | GarDiff†          | 0.8413 | 21.8914 | 0.0912  | 11.2894 | 0.0047  | -       | -       |
> | GarDiff (Ours)    | 0.8463 | 21.9647 | 0.0866  | 10.3414 | 0.0036  | -       | -       |
> | StableVITON*      | 0.8839 | 23.5965 | 0.0757  | 9.8694  | 0.0016  | 22.7463 | 0.0066 |
> | StableVITON†      | 0.8832 | 23.5586 | 0.0772  | 9.9520  | 0.0017  | 22.9052 | 0.0061 |
> | StableVITON (Ours)| 0.8859 | 23.7027 | 0.0746  | 9.7669  | 0.0009  | 22.5525 | **0.0040** |
> | IDM-VTON*         | 0.8440 | 20.1067 | 0.1193  | 11.5482 | 0.0050  | 27.1165 | 0.0125 |
> | IDM-VTON†         | 0.8441 | 20.1238 | 0.1201  | 12.1464 | 0.0058  | 26.4503 | 0.0105 |
> | IDM-VTON (Ours)   | 0.8477 | 20.4514 | 0.1183  | 11.5364 | 0.0048  | 24.3281 | 0.0086 |
> | OOTDiffusion*     | 0.8601 | 20.5861 | 0.0977  | 10.2954 | 0.0013  | 22.2065 | 0.0056 |
> | OOTDiffusion†     | 0.8564 | 20.6652 | 0.0968  | 10.3767 | 0.0022  | 22.2288 | 0.0058 |
> | OOTDiffusion (Ours)| 0.8583 | 21.3201 | 0.0953  | **9.5175** | 0.0009  | 23.0877 | 0.0052 |
> | ITA-MDT*          | 0.8760 | 23.5203 | 0.0764  | 9.7530  | 0.0025  | 23.0280 | 0.0073 |
> | ITA-MDT†          | 0.8813 | 23.5662 | 0.0748  | 10.1198 | 0.0029  | 22.3726 | 0.0072 |
> | ITA-MDT (Ours)    | 0.8820 | 23.5735 | 0.0737  | 9.5977  | 0.0024  | 22.0231 | 0.0059 |

---

> ### Author Response · Authors · 2025-11-22
> **Response to Reviewer Jd4t (2/4)**
>
> > - Notably, Poisson Blending operates remains a pixel-space post-hoc processing method. Since it modifies generated images after the fact without fully respecting the measurement, it cannot modify the generative trajectory or maintain consistency with the measurement during sampling. In contrast, ART-VITON directly integrates measurement constrains into the sampling trajectory through:
> >    - Latent-space alignment: Enforces measurement consistency directly in latent space, preventing fundamental inconsistencies at generation time.
> >    - Semantic coherence: Maintains alignment between try-on and non-try-on regions through data consistency optimization.
> >    - High-frequency preservation: Recovers VAE compression losses via frequency-domain correction, ensuring consistent visual quality across all regions.
> > - This comprehensive approach overcomes the intrinsic limitations of pixel-space post-processing.
> > - **Summary**: Post-hoc blending techniques attempt to smooth boundaries after generation, but cannot address the intrinsic latent-space inconsistencies responsible for artifacts. Our experiments demonstrate that effective artifact mitigation requires intervention during generation, supporting the validity of our inverse problem formulation.
>
> [1] Zeng, Jianhao, et al. "Cat-dm: Controllable accelerated virtual try-on with diffusion model." *Proceedings of the IEEE/CVF conference on computer vision and pattern recognition*. 2024.
>
> [2] Wan, Siqi, et al. "Improving virtual try-on with garment-focused diffusion models." *European Conference on Computer Vision*. Cham: Springer Nature Switzerland, 2024.
>
> [3] Kim, Jeongho, et al. "Stableviton: Learning semantic correspondence with latent diffusion model for virtual try-on." *Proceedings of the IEEE/CVF conference on computer vision and pattern recognition*. 2024.
>
> [4] Choi, Yisol, et al. "Improving diffusion models for authentic virtual try-on in the wild." *European Conference on Computer Vision*. Cham: Springer Nature Switzerland, 2024.
>
> [5] Xu, Yuhao, et al. "Ootdiffusion: Outfitting fusion based latent diffusion for controllable virtual try-on." *Proceedings of the AAAI Conference on Artificial Intelligence*. Vol. 39. No. 9. 2025.
>
> [6] Hong, Ji Woo, et al. "ITA-MDT: Image-Timestep-Adaptive Masked Diffusion Transformer Framework for Image-Based Virtual Try-On." *Proceedings of the Computer Vision and Pattern Recognition Conference*. 2025.

---

> ### Author Response · Authors · 2025-11-22
> **Response to Reviewer Jd4t (3/4)**
>
> **[W2] Furthermore, the proposed sampling method seems extremely complex. Each sampling step involves: (1) a VAE decode and encode (Eq. 5), (2) a latent-space optimization (Eq. 6), and (3) both a Fourier Transform and an Inverse Fourier Transform (Eq. 7). This process is almost certainly much slower than standard DDIM sampling or even other solvers like TReg. The authors should include a discussion of the model's inference time and computational overhead.**
>
> > - We have added computational efficiency analysis, including inference time and memory usage for all inverse solvers on VITON-HD using an NVIDIA RTX A5000 GPU, in **Table 2**. The baseline StableVITON uses standard DDIM sampling.
> > | Method               | SSIM ↑    | PSNR ↑    | LPIPS ↓  | FID ↓     | KID ↓   | **Inf.** (s) ↓ | **Mem.** (GB) ↓ |
> > |----------------------|-----------|-----------|----------|-----------|---------|------------|-------------|
> > | StableVITON (baseline) | 0.8839   | 23.5965   | 0.0757   | 9.8694    | 0.0016  | 9.117      | 7.66        |
> > | RePaint              | 0.8856    | 23.6635   | 0.0752   | 10.0829   | 0.0018  | 9.241      | 7.66        |
> > | MCG                  | 0.8855    | 23.6641   | 0.0752   | 10.085    | 0.0015  | 13.518     | 10.38       |
> > | DPS                  | 0.8851    | 23.6390   | 0.0749   | 9.9425    | 0.0014  | 13.573     | 10.38       |
> > | DreamSampler         | 0.8904    | **23.8984** | 0.0771   | 10.5143   | 0.0018  | 25.463     | 7.66        |
> > | FIG                  | 0.8851    | 23.6390   | 0.0749   | 9.9427    | 0.0014  | 13.389     | 10.38       |
> > | TReg                 | **0.8909** | 23.8205   | 0.0844   | 11.7467   | 0.0024  | 11.382     | 7.66        |
> > | Ours                 | 0.8859    | 23.7027   | **0.0746** | **9.7669** | **0.0009** | 12.101     | 7.66        |
> > - Key observations of ART-VITON:
> >    - Comparable runtime (12.1s) to TReg [1] (11.4s): only +0.7s, +6.1% overhead.
> >    - 2x faster than DreamSampler [2] (22.5s)
> >    - Similar or faster than gradient-based methods (MCG [3], DPS [4], FIG [5]: ~13.5s)
> >    - Lower memory (7.66 GB) than gradient-based optimization methods (~10.38 GB)
> >    - Moderate overhead, but significantly better performance consistency across metrics
> > - Efficiency analysis:
> >    - VAE encoding/decoding **(Eq. 5)** is required by other inverse solvers including TReg and DreamSampler, not unique to ART-VITON.
> >    - Latent-space optimization **(Eq. 6**) is grdient-free, adding negligible computation, thus memory-efficient.
> >    - Fourier-based frequency fusion **(Eq. 7)** constitutes the majority of added computation, but only 0.7s per image, enabling strong detail restoration without high cost.
> >    $$\\hat{\\mathbf{x}}\_{\\mathbf{y}} = \\mathbf{M} \\odot \\mathbf{y} + (1 - \\mathbf{M}) \\odot \\mathcal{D}(\\mathbf{z}) -(5)$$
> >    $$\\hat{\\mathbf{z}}\_0^{(t)}(\\bar \\alpha\_{t-1}) = \\bar \\alpha\_{t-1} \\hat{\\mathbf{z}}\_{\\mathbf{y}} + (1 - \\bar \\alpha\_{t-1})\\hat{\\mathbf{z}}\_0^{(t)} -(6)$$
> >    $$\\hat{\\mathbf{z}}\_0^{(t)}(\\bar \\alpha\_{t-1}) = \\mathbf{M}\odot (\\bar\\alpha\_{t-1}\\hat{\\mathbf{z}}\_{\\mathbf{y}}' + (1-\\bar\\alpha\_{t-1})\\hat{\\mathbf{z}}\_0^{(t)}) + (1-\\mathbf{M})\\odot \\hat{\\mathbf{z}}\_0^{(t)} - (7)$$
> > - While slightly slower than TReg due to Fourier fusion, ART-VITON maintains performance without trade-offs.
> > - Performance-Efficiency Balance:
> >    - Unlike existing methods that improve some aspects at the cost of others:
> >        - DreamSampler improves structural metrics (SSIM/PSNR) but severely harms perceptual similarity and FID/KID, while being slow (25.5s).
> >        - DPS/FIG improve FID but degrade SSIM/PSNR with runtime (~13.5s) and introduce boundary distortions.
> >        - ART-VITON consistently improves all metrics across paired and unpaired settings at moderate inference time (12.1s).
> > - Although ART-VITON performs additional operations per sampling step, the measured overhead is small and comparable to popular inverse solvers, while providing significantly more stable results without metric trade-offs.
>
> [1] Kim, Jeongsol, et al. "Regularization by Texts for Latent Diffusion Inverse Solvers." *The Thirteenth International Conference on Learning Representations*, 2025
>
> [2] Kim, Jeongsol, Geon Yeong Park, and Jong Chul Ye. "Dreamsampler: Unifying diffusion sampling and score distillation for image manipulation." *European Conference on Computer Vision*. Cham: Springer Nature Switzerland, 2024.
>
> [3] Chung, Hyungjin, et al. "Improving diffusion models for inverse problems using manifold constraints." *Advances in Neural Information Processing Systems* 35 (2022): 25683-25696.
> [4] Chung, Hyungjin, et al. "Diffusion posterior sampling for general noisy inverse problems." *arXiv preprint arXiv:2209.14687* (2022).
>
> [5] Yan, Yici, et al. "FIG: Flow with Interpolant Guidance for Linear Inverse Problems." *The Thirteenth International Conference on Learning Representations*.

---

> ### Author Response · Authors · 2025-11-22
> **Response to Reviewer Jd4t (4/4)**
>
> **[W3] Finally, Figure 2, as the main flowchart for the method, is overly dense and difficult to understand.**
>
> > - We have revised the **Figure 2 and caption** to improve interpretability.
> > - Key updates:
> >    1. Clear depiction of component **(A)**: We updated sub-figure of component (A) to explicitly show the role of the residual prior step and added a concise description:
> >    > *(A) Prior-based initialization places $\\mathbf{z}\_T$ on the data manifold, enabling valid posterior sampling for the inverse problem.*
> >    2. Visualization of mask semantics:
> >        - We added a binary mask $\\mathbf{M}$ overlay to **Figure 2(B)** to clearly illustrate the definition of mask regions:
> >            - Measurement regions (non-try-on, with value $1$): Face, hair, background—areas in the original image that should be preserved.
> >            - Masked regions (try-on, with value $0$): Areas where the clothing needs to be generated.
> >        - This clarifies why our definition of mask **$\\mathbf{M}$** reverses the convention in prior VITON work (where **$\\mathbf{M}$** denotes only the inpainting garment region), aligning it with inverse problem notation where the observed signal represents the measurements.
> > - The updated **Figure 2** now enhance the interpretability and clearly communicate the role of each component in our method.

---

> ### Author Response · Authors · 2025-11-25
> **[Updated Response to W2] Computational Cost Analysis with FLOPs**
>
> **[W2] Furthermore, the proposed sampling method seems extremely complex. Each sampling step involves: (1) a VAE decode and encode (Eq. 5), (2) a latent-space optimization (Eq. 6), and (3) both a Fourier Transform and an Inverse Fourier Transform (Eq. 7). This process is almost certainly much slower than standard DDIM sampling or even other solvers like TReg. The authors should include a discussion of the model's inference time and computational overhead.**
>
> > - We have added FLOPs (floating-point operations) measurements to **Table 2** to provide a complete computational efficiency analysis:
> > | Method | SSIM ↑ | PSNR ↑ | LPIPS ↓ | FID ↓ | KID ↓ | FLOPs (T) ↓ | Inf. (s) ↓ | Mem. (GB) ↓ |
> > |--------|--------|--------|---------|-------|-------|-------------|------------|--------------|
> > | StableVITON (baseline) | 0.8839 | 23.5965 | 0.0757 | 9.8694 | 0.0016 | 86.178 | 9.117 | 7.66 |
> > | RePaint | 0.8856 | 23.6635 | 0.0752 | 10.0829 | 0.0018 | 87.864 | 9.241 | 7.66 |
> > | MCG | 0.8855 | 23.6641 | 0.0752 | 10.085 | 0.0015 | 259.87 | 13.518 | 10.38 |
> > | DPS | 0.8851 | 23.6390 | 0.0749 | 9.9425 | 0.0014 | 259.87 | 13.573 | 10.38 |
> > | DreamSampler | 0.8904 | 23.8984 | 0.0771 | 10.5143 | 0.0018 | 267.22 | 25.463 | 7.66 |
> > | FIG | 0.8851 | 23.6390 | 0.0749 | 9.9427 | 0.0014 | 259.87 | 13.389 | 10.38 |
> > | TReg | 0.8909 | 23.8205 | 0.0844 | 11.7467 | 0.0024 | 130.63 | 11.382 | 7.66 |
> > | Ours | 0.8859 | 23.7027 | 0.0746 | 9.7669 | 0.0009 | 130.63 | 12.101 | 7.66 |
> > -  ART-VITON computational cost (130.63 TFLOPs):
> >    - Identical to TReg (130.63 TFLOPs)
> >    * 1.5× higher than baseline (86.178 TFLOPs)
> >    * 2× lower than gradient-based methods (MCG, DPS, FIG: ~260 TFLOPs)
> >    * 2× lower than DreamSampler (267.22 TFLOPs)
> > * Breakdown of additional FLOPs:
> >    * The primary overhead arises from the repeated VAE encoding/decoding steps.
> >    * Although inherently expensive, these iterations are essential for effective pixel optimization.
> > * Efficiency-Quality Trade-off:
> >    * Gradient-based methods require 2× FLOPs but still fail to improve unpaired metrics
> >    * DreamSampler has highest FLOPs (267.22 TFLOPs) yet degrades perceptual quality
> >    * ART-VITON achieves best overall performance with moderate FLOPs, demonstrating efficient use of computation
> > - The FLOPs measurements confirm that ART-VITON maintains reasonable computational cost while being the only method to consistently improve all metrics without trade-offs.

---

### Official Review · Reviewer_LxsW · 2025-11-04

**Soundness:** 3
**Presentation:** 2
**Contribution:** 3
**Rating:** 4
**Confidence:** 4

**Summary:**

This paper considers virtual try-on. The focus of this paper is on the improvement in generation of the non-garment regions that should be preserved. The proposed approach adapts the intermediate generated image so that the inpainted and existing regions are mode continuous. At regular intervals during the diffusion process, the predicted latent $\hat{z}_0^{(t)}$ image is decoded and fused with the masked model image (the masked region comes from the decoded image, the non-masked region comes from the conditioning image). This image is again encoded into a latent $\hat{z}_y$. To insure consistency, interpolation between $\hat{z}_y$ and  $\hat{z}_0^{(t)}$ is done and a further frequency correction is done to preserve detail. This process is interchangeably done with regular diffusion steps for a more stable diffusion process.

**Strengths:**

[S1] The proposed approach is inference-time, and may therefore be used to improve existing VTON pipelines.

[S2] Good quantitative and qualitative results.

**Weaknesses:**

[W1] Missing ablations for important choices in the method (e.g. the strength of interpolation in equations 6 and 7, the number of steps for C).

[W2] Prior-based initialization is not novel and I am not completely certain how it relates to reformulating VTON as inverse problem and equation 4.

[W3] I thought that the paper is badly written and difficult to follow. See questions. I also thought some of the naming/symbol conventions made the paper more difficult to follow. For example, naming the masked area as a measurement. A measurement would imply some sort of injection of physical constraints into to model. Similarly, tradinally in VTOM paper M indicates the masked region that should be inpainted, but here it is reversed.

**Questions:**

[Q1] What is being shown by the Figure 3? Is the determination of artefacts quantitative and measurable or just qualitative? How is failure/success determined?

[Q2] Some elements of the method are not completely clear, e.g. how is $\hat{z}'_y$ obtained?

---

> ### Author Response · Authors · 2025-11-22
> **Response to Reviewer LxsW (1/6)**
>
> **[W1] Missing ablations for important choices in the method (e.g. the strength of interpolation in equations 6 and 7, the number of steps for C).**
>
> > - We provide comprehensive ablation studies in **Appendix 4: Additional Ablation Study**, specifically addressing these components. Below we summarize the key findings.
> > 1. Interpolation strength in **Eq. 6 (Table 7)**
> >     - **Equation 6** performs data-consistency optimization by balancing the latent $\\hat{\\mathbf{z}}_{\\mathbf{y}}$ optimized from the measurement with the reference-informed latent $\\hat{\\mathbf{z}}_0^{(t)}$ from Tweedie estimation.
> >  $$\\min\_{\\mathbf{z}} \\left\\| \\frac{\\hat{\\mathbf{z}}\_0^{(t)} - \\mathcal{E}(\\mathcal{D}(\\mathbf{z}))}{2\\sigma\_{\\mathcal{E}}^2} \\right\\|\_2^2, \\hat{\\mathbf{z}}\_0^{(t)}(\\bar \\alpha\_{t-1}) = \\bar \\alpha\_{t-1} \\hat{\\mathbf{z}}\_{\\mathbf{y}} + (1 - \\bar \\alpha\_{t-1})\\hat {\\mathbf{z}}\_0^{(t)} -(6) $$
> >     - This update can be expressed as a quadratic minimization problem, where the weights $(\lambda_{\text{curr}}, \lambda_{\text{ref}})$ control the trust between the two latents.
> >  $$\\hat{\\mathbf{z}}\_{0}^{(t)}(\\lambda\_{\text{curr}}) = \\arg\\min\_{\mathbf{z}}  \\lambda\_{\text{curr}} \\left\\| \\mathbf{z} - \\hat{\\mathbf{z}}\_{\mathbf{y}}\\right\\|\_2^2 + \\lambda\_{\text{ref}} \\left\\|\\mathbf{z} - \\hat{\\mathbf{z}}\_{0}^{(t)}\\right\\|\_2^2,$$
> >     - which has the closed-form solution:
> >  $$\\mathbf{z} = \\frac{\\lambda\_{\text{curr}}}{\\lambda\_{\text{curr}} + \\lambda\_{\text{ref}}} \\hat{\\mathbf{z}}\_{\mathbf{y}} + \\frac{\\lambda\_{\text{ref}}}{\\lambda\_{\text{curr}} + \\lambda\_{\text{ref}}} \\hat{\\mathbf{z}}\_{0}^{(t)}.$$
> >     - A key insight here is that we adopt a time-dependent weighting scheme instead of a fixed one, where $\\bar \\alpha\_{t-1}$ gradually increases and $(1-\\bar \\alpha\_{t-1})$ decreases as denoising progresses. Early in sampling (near $t\\approx T{=}999$), $\\hat{\\mathbf{z}}\_0^{(t)}$ is still far from the true latent $\\mathbf{z}_{\\mathbf{y}}$ **(red line in Figure. 2 (B-2))**, so forcing strong alignment with $\\hat{\\mathbf{z}}\_{\\mathbf{y}}$ causes semantic mismatches between masked and measured regions. Later in sampling ($t\\rightarrow 0$), however, $\\hat{\\mathbf{z}}\_0^{(t)}$ becomes progressively closer to $\\mathbf{z}\_{\mathbf{y}}$, making it necessary to increase the weight on $\\hat{\\mathbf{z}}\_{\\mathbf{y}}$. This adaptive strategy preserves consistency with the diffusion dynamics and ensures stable trajectory alignment throughout the process.
> >     - We evaluate multiple weighting strategies $(\\lambda\_{\\text{curr}}, \\lambda\_{\\text{ref}})$ for balancing the optimized latent $\\hat{\\mathbf{z}}_{\\mathbf{y}}$ and reference-informed latent $\\hat{\\mathbf{z}}\_0^{(t)}$, as shown in **Table 7**. We compared early-biased and late-biased weighting, fixed weights, with our time-dependent weighting scheme $\\lambda\_\\text{curr} = \\frac{\\lambda\_{\\text{ref}}\\bar\\alpha\_{t-1}}{1-\\bar\\alpha\_{t-1}}$. Our adaptive weighting achieves the the best unpaired metrics (FID, KID) and competitive paired metrics (SSIM, PSNR, LPIPS), demonstrating that time-aligned balancing between the two latents is essential for stable and artifact-free synthesis.
> >   | $\\lambda\_{\\text{curr}}$ | $\\lambda\_{\\text{ref}}$ | SSIM ↑ | PSNR ↑ | LPIPS ↓ | FID ↓ | KID ↓ |
>      |------------------------|------------------------|--------|--------|---------|--------|--------|
>      | 1                    | 0                    | 0.8677 | 22.5991 | 0.1623 | 20.1817 | 0.0088 |
>       | 0                    | 1                    | 0.8552 | 23.1475 | 0.0833 | 10.4362 | 0.0014 |
>      | $\\lambda\_{\\text{ref}}$ | $\\lambda\_{\\text{curr}}$ | 0.8862 | 23.7223 | 0.0748 | 9.8565 | 0.0010 |
>    | $\\frac{\\lambda\_{\\text{ref}} (1-\\bar\\alpha\_{t-1})}{\\bar\\alpha\_{t-1}}$ | $-$ | **0.8864** | **23.7327** | 0.0749 | 9.8938 | 0.0012 |
>     | $\\frac{\\lambda\_{\\text{ref}}\\bar\\alpha\_{t-1}}{1-\\bar\\alpha\_{t-1}}$ | $-$ | 0.8859 | 23.7027 | **0.0746** | **9.7669** | **0.0009** |

---

> ### Author Response · Authors · 2025-11-22
> **Response to Reviewer LxsW (2/6)**
>
> > 2. Number of standard denoising steps $N$ **(Table 8)**:
> >     - Component **(C)** re-anchors and stabilizes the sampling trajectory to the manifold by periodically applying standard denoising after the measurement-guided optimization step. $N$ denotes the number of standard denoising steps applied between each optimization step. We evaluate $N \\in \\{1, 2, 3,5, 10, 25, 50\\}$ (see **Table 8**):
> >     - $N=1$ applies measurement-guided optimization too frequently. This yields strong paired metrics because the measurement is enforced aggressively, but the trajectory does not have enough time to realign with the diffusion manifold,  leading to degraded unpaired performance.
> >    - $N \\ge 25$ applies optimization too infrequently, thus the sampling trajectory is dominated by standard denoising. It prevents effective injection of measurement information and lowers performance across all metrics.
> >    - $N = 2$ strikes the optimal balance, achieving stable trajectories while still incorporating measurement constraints consistently. This configuration achieves the best joint performance across paired and unpaired settings.
> > | (C) Std. denoising | SSIM ↑ | PSNR ↑ | LPIPS ↓ | FID ↓ | KID ↓ |
> |-------------------|--------|--------|---------|-------|-------|
> | $N = 1$             | **0.8860** | **23.7041** | 0.0748 | 9.8309 | 0.0013 |
> | $N = 2$             | 0.8859 | 23.7027 | **0.0746** | **9.7669** | **0.0009** |
> | $N = 3$             | 0.8858 | 23.6891 | **0.0746** | 9.7841 | 0.0014 |
> | $N = 5$             | 0.8857 | 23.6780 | **0.0746** | 9.7687 | 0.0013 |
> | $N = 10$            | 0.8855 | 23.6602 | **0.0746** | 9.8106 | 0.0016 |
> | $N = 25$            | 0.8852 | 23.6387 | 0.0747 | 9.9489 | 0.0012 |
> | $N = 50$            | 0.8552 | 23.1475 | 0.0833 | 10.4362 | 0.0014 |

---

> ### Author Response · Authors · 2025-11-22
> **Response to Reviewer LxsW (3/6)**
>
> **[W2] Prior-based initialization is not novel and I am not completely certain how it relates to reformulating VTON as inverse problem and equation 4.**
>
> > - Novelty and Distinction from Prior Work: While prior initialization techniques exist in diffusion literature, their objectives and mechanisms differ substantially from ours.
> >     - Existing methods:
> >        - PASD [1], SeeSR [2]: Mix “external” low-quality inputs with noise → requires degraded images that are unavailable in VITON
> >        - DCI-VTON [3]: Mixes warped garments into noise → depends on an additional warping module and is not model-agnostic
> >        - Common limitations: None of these works address the training-inference mismatch caused by: training with residual signals at $T=999$ and inference starting from pure noise at $T=981$ (typical in DDIM-based VITON). This mismatch is especially problematic when VITON is framed as an inverse problem, because misaligned initial latents degrade both semantic consistency and measurement preservation.
> >    - Our approach:
> >        - Start from pure Gaussian $\\mathbf{z}\_{999}$, Apply $1$ DDPM step and obtain residual prior $\\mathbf{z}\_{998}$ that contains the correct residual structure
> >        - Model-agnostic & zero-cost: No additional modules, preprocessing, or external data required
> >        - Leverages existing diffusion prior knowledge to resolve train-test misalignment in a single step
> > - Direct Connection to Inverse Problem Formulation **(Eq. 4)**
> >     - Our initialization is not an isolated trick - it is required to make the inverse-problem formulation in **Eq. 4** valid during inference:
> >    $$p(\\mathbf{x} | \\mathbf{y}, \\hat{\\mathbf{z}}\_0^{(t)}) \\propto p(\\hat{\\mathbf{z}}\_0^{(t)} | \\mathcal{D}(\\mathbf{z}), \\mathbf{y}) \\cdot p(\\mathbf{y} | \\mathcal{D}(\\mathbf{z})) - (4)$$
> >    1. Manifold-constrained starting point:
> >        - For posterior sampling $p(\\mathbf{x} | \\mathbf{y}, \\hat{\\mathbf{z}}\_0^{(t)})$ in **Eq. 4**, it requires $\\mathbf{z}\_T$ to lie on the noisy data manifold $\\mathcal{M}\_T$ the model was trained on. Standard inference, starting from pure Gaussian noise at reduced timesteps, creates an off-manifold initialization, whereas our prior-based initialization places $\\mathbf{z}\_T$ on the correct manifold $\\mathcal{M}\_T$ by reconstructing the learned noise statistics exactly as in training, thus enabling accurate estimation of both posterior terms: semantic plausibility $p(\\hat{\\mathbf{z}}\_0^{(t)} | \\mathcal{D}(\\mathbf{z}), \\mathbf{y})$ and measurement constraint $p(\\mathbf{y} | \\mathcal{D}(\\mathbf{z}))$.
> >    2. Error propagation minimization:
> >        - If $\\mathbf{z}\_T$ is off-manifold, semantic plausibility term $p(\\hat{\\mathbf{z}}\_0^{(t)} | \\mathcal{D}(\\mathbf{z}), \\mathbf{y})$ becomes unstable early in the trajectory, and measurement constraint term $p(\\mathbf{y} | \\mathcal{D}(\\mathbf{z}))$ degrades due to region mismatches that accumulate across time steps. Our on-manifold initialization mitigates this by ensuring diffusion steps begin from the correct manifold structure, reducing drift from the very first iteration.
> >    3. Universal improvement across baselines **(Table 1)**:
> >        - Our initialization improves various baselines (DCI-VTON [3], GarDiff [4], StableVITON [5]) with $T=999$ prior initialization. This architecture-agnostic gain supports our claim that:
> >        > *Proper initialization is a fundamental requirement of the inverse problem formulation, not an architecture-specific heuristic.*
> > **Summary**: Prior-based initialization in ART-VITON is not a heuristic or preprocessing trick. It is a necessary foundation for: ensuring $\\mathbf{z}\_T$ lies on the correct noisy manifold, enabling valid posterior sampling under **Eq. 4**, and stabilizing both semantic and measurement-related terms in the inverse-problem formulation.
>
> [1] Yang, Tao, et al. "Pixel-aware stable diffusion for realistic image super-resolution and personalized stylization." *European conference on computer vision*. Cham: Springer Nature Switzerland, 2024.
>
> [2] Wu, Rongyuan, et al. "Seesr: Towards semantics-aware real-world image super-resolution." *Proceedings of the IEEE/CVF conference on computer vision and pattern recognition*. 2024.
>
> [3] Gou, Junhong, et al. "Taming the power of diffusion models for high-quality virtual try-on with appearance flow." *Proceedings of the 31st ACM International Conference on Multimedia*. 2023.
>
> [4] Wan, Siqi, et al. "Improving virtual try-on with garment-focused diffusion models." *European Conference on Computer Vision*. Cham: Springer Nature Switzerland, 2024.
>
> [5] Kim, Jeongho, et al. "Stableviton: Learning semantic correspondence with latent diffusion model for virtual try-on." *Proceedings of the IEEE/CVF conference on computer vision and pattern recognition*. 2024.

---

> ### Author Response · Authors · 2025-11-22
> **Response to Reviewer LxsW (4/6)**
>
> **[W3] I thought that the paper is badly written and difficult to follow. See questions. I also thought some of the naming/symbol conventions made the paper more difficult to follow. For example, naming the masked area as a measurement. A measurement would imply some sort of injection of physical constraints into to model. Similarly, tradinally in VTOM paper M indicates the masked region that should be inpainted, but here it is reversed**
>
> > - Use of the term “measurement”: Our terminology follows standard inverse problem literature (e.g., [1, 2, 3]), where $\\mathbf{y} = \\mathcal{A}\\mathbf{x}$ represents observed constraints that must remain unchanged during reconstruction. In VITON, the **non-try-on regions** (e.g., face, background) are indeed observed variables and must be preserved exactly. Therefore, referring to these pixels as “measurements” is mathematically appropriate in the context of our formulation - directly corresponds to $\\mathbf{y}$. We prevent early misunderstanding by labeling the clothing-agnostic map as the measurement directly inside **Figure 1**, explicitly showing that the measurement corresponds to the preserved non-try-on region.
> > - Mask notation $\\mathbf{M}$: We agree that our mask definition differs from typical VITON papers where $\\mathbf{M}$ denotes the try-on-region (i.e., clothing alignment) requiring synthesis. In our inverse-problem formulation, the conceptual focus is different. We emphasize **preserving non-try-on region** (the observed measurement), thus our mask $\\mathbf{M}$ naturally denotes the preserved region. To reduce confusion, we made the revisions: added a clear visualization of mask $\\mathbf{M}$ in **Figure 2(B)**.
>
> [1] Chung, Hyungjin, et al. "Diffusion posterior sampling for general noisy inverse problems." *arXiv preprint arXiv:2209.14687* (2022).
>
> [2] Kim, Jeongsol, Geon Yeong Park, and Jong Chul Ye. "Dreamsampler: Unifying diffusion sampling and score distillation for image manipulation." *European Conference on Computer Vision*. Cham: Springer Nature Switzerland, 2024.
>
> [3] Kim, Jeongsol, et al. "Regularization by Texts for Latent Diffusion Inverse Solvers." *The Thirteenth International Conference on Learning Representations*, 2025

---

> ### Author Response · Authors · 2025-11-22
> **Response to Reviewer LxsW (5/6)**
>
> **[Q1] What is being shown by the Figure 3? Is the determination of artefacts quantitative and measurable or just qualitative? How is failure/success determined?**
>
> > - Purpose of **Figure 3**:
> >    - **Figure 3** is intended as a **qualitative** diagnostic tool that visualizes the characteristic failure modes of different categories of inverse solvers. Goal of this figure is not to produce a binary success/failure judgement, but to highlight **where and how artifacts appear,** especially at boundaries between try-on and non-try-on regions - precisely the challenge our method aims to address. Standard image crops often fail to reveal subtle boundary inconsistencies; thus we use gradient magnitude heatmaps to make spatial discontinuities clearly visible. This quantifies abrupt changes at boundaries and represents them using color, highlighting the regions most affected by artifacts.
> > - Method-specific issues highlighted in **Figure 3(a)** and **Figure 3(b)**
> >    - **Figure 3(a)** Hybrid stochastic methods (DreamSampler [1], TReg [2]): Noise injection mitigates boundary artifacts but introduces high-frequency loss during VAE re-encoding (red circles indicate blurred textures and missing fine details).
> >    - **Figure 3(b)** Hard constraint & Progressive methods (RePaint [3], MCG [4], DPS [5], FIG [6]): These methods do not fully satisfy measurement constraints during sampling and rely on post-hoc replacement, producing strong boundary discontinuities.
> >    - The figure also illustrates each method’s limitations on additional samples to emphasize their consistent behavior. We include multiple examples in **Figure 11** to demonstrate that these behaviors are consistent, not sample-specific.
> > - Quantitative vs. qualitative evaluation
> >    - Artifact perception is inherently subjective and localized; global metrics (PSNR/SSIM/FID/KID) often fail to reflect fine-grained boundary inconsistencies or localized artifacts.
> >    - Thus, we provide both **Table 2 (quantitative) and Figure 3 (qualitative)** to comprehensively evaluate each method from multiple perspectives. In response to the reviewer’s comments, we have also revised the **caption of Figure 3** and the main text (**Section. 5.2**) to make this purpose clearer.
> >    > Caption: *“Qualitative comparison across inverse solver categories: (a) Hybrid stochastic methods (DreamSampler, TReg) reduce visible artifacts but lose high-frequency details. (b) Hard constraint (RePaint, MCG) and progressive methods (DPS, FIG) rely on post-hoc replacement, leading to boundary artifacts. Our method mitigates both issues, preserving fine details and enhancing visual coherence across all regions.”*
>
> [1] Kim, Jeongsol, Geon Yeong Park, and Jong Chul Ye. "Dreamsampler: Unifying diffusion sampling and score distillation for image manipulation." *European Conference on Computer Vision*. Cham: Springer Nature Switzerland, 2024.
>
> [2] Kim, Jeongsol, et al. "Regularization by Texts for Latent Diffusion Inverse Solvers." *The Thirteenth International Conference on Learning Representations*, 2025
>
> [3] Lugmayr, Andreas, et al. "Repaint: Inpainting using denoising diffusion probabilistic models." *Proceedings of the IEEE/CVF conference on computer vision and pattern recognition*. 2022.
>
> [4] Chung, Hyungjin, et al. "Improving diffusion models for inverse problems using manifold constraints." *Advances in Neural Information Processing Systems* 35 (2022): 25683-25696.
>
> [5] Chung, Hyungjin, et al. "Diffusion posterior sampling for general noisy inverse problems." *arXiv preprint arXiv:2209.14687* (2022).
>
> [6] Yan, Yici, et al. "FIG: Flow with Interpolant Guidance for Linear Inverse Problems." *The Thirteenth International Conference on Learning Representations*.

---

> ### Author Response · Authors · 2025-11-22
> **Response to Reviewer LxsW (6/6)**
>
> **[Q2] Some elements of the method are not completely clear, e.g. how is $\\hat{\\mathbf{z}}\_{\\mathbf{y}}'$obtained?**
>
> > - We agree that the original manuscript did not sufficiently clarify the construction of $\\hat{\\mathbf{z}}\_{\\mathbf{y}}'$. We have now added a detailed explanation and equation in **L265** of the revised manuscript: it is obtained by fusing two latents in the frequency domain
> > $$\\hat{\\mathbf{z}}\_\\mathbf{y}' = \\hat{\\mathbf{z}}\_\\mathbf{y}^{\\text{low}} + \\hat{\\mathbf{z}}\_0^{(t),\\text{high}}$$
> > - Step-by-step process:
> >    1. Fourier transform: Apply a channel-wise Fourier transform to both $\\hat{\\mathbf{z}}\_{\\mathbf{y}}$ and  $\\hat{\\mathbf{z}}\_0^{(t)}$.
> >    2. Frequency component separation:
> >        - Extract low-frequency components $\\hat{\\mathbf{z}}\_{\\mathbf{y}}^{\\text{low}}$ from measurement-consistent latent $\\hat{\\mathbf{z}}\_{\\mathbf{y}}$.
> >        - Extract high-frequency components $\\hat{\\mathbf{z}}\_{0}^{\\text{(t), high}}$ from reference-informed latent $\\hat{\\mathbf{z}}\_0^{(t)}$.
> >    3. Frequency-domain fusion: Combine semantic structure (the optimized low-frequency) with fine texture details (the reference high-frequency).
>  >   4. Inverse Fourier transform: Convert the fused frequency representation back to latent space to obtain $\\hat{\\mathbf{z}}\_{\\mathbf{y}}'$.
> > - VAE re-encoding in **Eq. (5)** is known to attenuate high-frequency details [1, 2, 3]. Thus, directly interpolating $\\hat{\\mathbf{z}}\_{\\mathbf{y}}$ as in **Eq. (6)** would propagate high-frequency loss (e.g., blur) across regions and degrade garment realism. Our frequency-domain fusion restores sharp details while retaining measurement consistency.
> > - As illustrated in **Method Overview (Figure 2, steps 4-5)**, we keep values from $\\hat{\\mathbf{z}}\_0^{(t)}$ unchanged in masked regions (try-on), and apply frequency-corrected $\\hat{\\mathbf{z}}\_{\\mathbf{y}}'$ to measurement regions (non-try-on). This maintains the optimized low-frequency semantic layout while recovering textures and edges from the reference-informed latent $\\hat{\\mathbf{z}}\_0^{(t)}$. It ensures coherent textures and artifact-free transitions between regions.
>
> [1] Zhang, Jinjin, et al. "Diffusion-4k: Ultra-high-resolution image synthesis with latent diffusion models." *Proceedings of the Computer Vision and Pattern Recognition Conference*. 2025.
>
> [2] Novitskiy, Lev, et al. "VIVAT: Virtuous Improving VAE Training through Artifact Mitigation." *arXiv preprint arXiv:2506.07863* (2025).
>
> [3] Almog, Gal, Ariel Shamir, and Ohad Fried. "REED‐VAE: RE‐Encode Decode Training for Iterative Image Editing with Diffusion Models." *Computer Graphics Forum*. 2025.

---

### Author Response · Authors · 2025-12-01
**To all Reviewers**

We sincerely thank all reviewers for their time and valuable feedback. **ART-VITON** is a measurement-guided latent diffusion framework that reformulates virtual try-on as a linear inverse problem to effectively mitigate boundary artifacts between try-on and non-try-on regions.  As noted by the reviewers, it can be easily applied at inference time to existing VTON pipelines and achieves consistent quantitative and qualitative improvements through clear problem formulation and thorough validation.

In response to the reviewers' questions and feedback, we summarize the revisions and additional experiments conducted during the rebuttal period as follows:

**Additional Experiments**
> 1. Comprehensive ablation studies on key hyperparameters *(Reviewer LxsW W1)*
>     - Ablation on interpolation strength in **Eq. 6 (Table 7 in Appendix)**: Evaluated multiple weighting strategies (early-biased, late-biased, fixed, time-dependent) demonstrating that time-dependent weighting achieves optimal balance across all metrics.
>     - Ablation on standard denoising frequency $N$ **(Table 8 in Appendix)**: Analyzed $N\\in{1,2,3,5,10,25,50}$, confirming $N=2$ provides the best trade-off between trajectory stability and measurement preservation.
> 2. Comparison with Poisson Blending *(Reviewer Jd4t W1)*
>     - Added quantitative comparison **(Table 3)** and qualitative analysis **(Figure 9)** against Poisson Blending used in CAT-DM [1].
Results show Poisson Blending degrades unpaired metrics (FID, KID) for most baselines, as gradient-domain smoothing cannot resolve latent-space semantic inconsistencies.
3. Computational efficiency analysis *(Reviewer Jd4t W2, CHdC W1)*
    - Added inference time, memory usage, and FLOPs measurements to **Table 2**.
    - ART-VITON achieves 130.63 TFLOPs (identical to TReg [2]), 12.1s inference time (+6.1% overhead vs TReg), and 7.66 GB memory—2× more efficient than gradient-based methods while achieving consistent improvements across all metrics.
 4. Extended baseline comparisons with recent VITON methods *(Reviewer CHdC W3, JK6a W1)*
     - Added three recent architectures: IDM-VTON [3] (SDXL), OOTDiffusion [4] (SD v1.5), and ITA-MDT [5] (Masked Diffusion Transformer).
     - Results in **Table 3** and **Figures 4, 5, 12-15** demonstrate consistent improvements across diverse architectures, validating the model-agnostic nature of our approach.
 5. Extended timestep analysis *(Reviewer CHdC W1)*
     - Added **Figure 8 (Appendix)** showing artifact maps across nine intermediate timesteps, demonstrating that ART-VITON maintains consistent trajectories while baseline inverse solvers (TReg [2]) exhibit progressive semantic drift.

**Clarifications**
> 1. Prior-based initialization and inverse problem connection *(Reviewer LxsW W2)*
>     - Distinguished our method from PASD/SeeSR/DCI-VTON [6, 7, 8] and clarified that proper initialization aligns $\\mathbf{z}\_{T}$ with the correct noisy manifold for stable posterior sampling.
> 2. Terminology and notation *(Reviewer LxsW W3)*
>     - Specified that “measurement” follows standard inverse problem terminology; added mask $\\mathbf{M}$ visualization in **Figure 2(B)** for clarity.
> 3. Purpose of **Figure 3** *(Reviewer LxsW Q1)*
>    - Clarified that **Figure 3** functions as a qualitative diagnostic, illustrating (a) high-frequency loss in hybrid stochastic methods and (b) boundary artifacts in hard/progressive solvers.
> 4. Revision of overclaimed statements *(Reviewer CHdC W2)*
>    - Replaced “artifact-free” with more moderate descriptions and added explicit limitations regarding full artifact removal.
> 5. High-frequency correction mechanism *(Reviewer JK6a W2)*
>    - Clarified that the contribution is global high-frequency restoration via frequency-domain fusion, not masking; explained mitigation of VAE re-encoding degradation through selective component fusion.

---

> ### Author Response · Authors · 2025-12-01
> **References**
>
> [1] Zeng, Jianhao, et al. "Cat-dm: Controllable accelerated virtual try-on with diffusion model." Proceedings of the IEEE/CVF conference on computer vision and pattern recognition. 2024.
>
> [2] Kim, Jeongsol, et al. "Regularization by Texts for Latent Diffusion Inverse Solvers." The Thirteenth International Conference on Learning Representations, 2025
>
> [3] Choi, Yisol, et al. "Improving diffusion models for authentic virtual try-on in the wild." European Conference on Computer Vision. Cham: Springer Nature Switzerland, 2024.
>
> [4] Xu, Yuhao, et al. "Ootdiffusion: Outfitting fusion based latent diffusion for controllable virtual try-on." Proceedings of the AAAI Conference on Artificial Intelligence. Vol. 39. No. 9. 2025.
>
> [5] Hong, Ji Woo, et al. "ITA-MDT: Image-Timestep-Adaptive Masked Diffusion Transformer Framework for Image-Based Virtual Try-On." Proceedings of the Computer Vision and Pattern Recognition Conference. 2025.[1] Yang, Tao, et al. "Pixel-aware stable diffusion for realistic image super-resolution and personalized stylization." European conference on computer vision. Cham: Springer Nature Switzerland, 2024.
>
> [6] Wu, Rongyuan, et al. "Seesr: Towards semantics-aware real-world image super-resolution." Proceedings of the IEEE/CVF conference on computer vision and pattern recognition. 2024.
>
> [7] Gou, Junhong, et al. "Taming the power of diffusion models for high-quality virtual try-on with appearance flow." Proceedings of the 31st ACM International Conference on Multimedia. 2023.

---

### Note · Authors · 2025-12-26

I have read and agree with the venue's withdrawal policy on behalf of myself and my co-authors.